# Germinal center B cell development has distinctly regulated stages completed by disengagement from T cell help

Ting-ting Zhang[1,2]*, David G Gonzalez[1,2], Christine M Cote[2], Steven M Kerfoot[3], Shaoli Deng[4], Yuqing Cheng[5], Masaki Magari[6], Ann M Haberman[1,2]*

[1]Department of Laboratory Medicine, Yale University School of Medicine, New Haven, United States; [2]Department of Immunobiology, Yale School of Medicine, New Haven, United States; [3]Department of Microbiology and Immunology, Western University, London, Canada; [4]Third Military Medical University, Chongqing, China; [5]Nanjing Normal University, Nanjing, China; [6]Department of Medical Bioengineering, Okayama University, Okayama, Japan

**Abstract** To reconcile conflicting reports on the role of CD40 signaling in germinal center (GC) formation, we examined the earliest stages of murine GC B cell differentiation. Peri-follicular GC precursors first expressed intermediate levels of BCL6 while co-expressing the transcription factors RelB and IRF4, the latter known to repress Bcl6 transcription. Transition of GC precursors to the BCL6$^{hi}$ follicular state was associated with cell division, although the number of required cell divisions was immunogen dose dependent. Potentiating T cell help or CD40 signaling in these GC precursors actively repressed GC B cell maturation and diverted their fate towards plasmablast differentiation, whereas depletion of CD4+ T cells promoted this initial transition. Thus while CD40 signaling in B cells is necessary to generate the immediate precursors of GC B cells, transition to the BCL6$^{hi}$ follicular state is promoted by a regional and transient diminution of T cell help.

*For correspondence: tingting. zhang@yale.edu (T-tZ); ann. haberman@yale.edu (AMH)

**Competing interests:** The authors declare that no competing interests exist.

## Introduction

Germinal centers (GCs) are specialized structures that form within B cell follicles in response to immunization or pathogen exposure. Within defined anatomical niches of mature GCs, antigen specific GC B cells intermittently engage cognate T follicular helper cells (T$_{fh}$). Competitive GC B cells are instructed to undergo further rounds of proliferation, and ultimately differentiate into memory B cells and long-term high-affinity plasma cells (*De Silva and Klein, 2015*). In contrast to mature GCs, the initial formation of GC founder/precursor cells remains poorly understood. The initiation of GCs is regulated by a series of dynamic cellular events. After antigen binding and B cell receptor (BCR) signaling, activated follicular B cells migrate to the T/B border and interfollicular (IF) regions. At the periphery of follicles, responding B and T cells engage, separate and re-form contacts with new cognate partners over the course of several days (*Kerfoot et al., 2011*). Initial B cell commitment to the GC lineage and divergence from the short-term antibody secreting cell (ASC) path occurs during this time period (*Kerfoot et al., 2011*). Intermediate levels of BCL6 have been observed in some activated B cells prior to follicular re-entry and GC seeding (*Kerfoot et al., 2011*; *Kitano et al., 2011*). Interestingly, Tfh migration into the follicle interior precedes that of GC committed B cells (*Kerfoot et al., 2011*).

Multiple mechanisms have been proposed to influence the mode of differentiation of activated B cells to either the GC or ASC lineages. In vitro culture studies have suggested that B cell fate decisions are a stochastic and B cell autonomous process (*Hasbold et al., 2004*, *1998*; *Hodgkin et al.,*

*1996*). Other studies have instead suggested that either the initial BCR signal strength (*O'Connor et al., 2006*; *Paus et al., 2006*) or competition for antigen and its presentation to cognate Th cells (*Schwickert et al., 2011*) are deciding factors. B cell lineage commitment is orchestrated by opposing transcriptional networks thought to be mutually antagonistic. The transcriptional repressor BCL6 is essential for GC B cell development (*De Silva and Klein, 2015*; *Dent et al., 1997*; *Huang et al., 2014*; *Kitano et al., 2011*) and represses the transcription factors IRF4 and Blimp1, preventing ASC formation (*Basso and Dalla-Favera, 2012*). By contrast, high levels of IRF4 or induction of Blimp1 represses transcription of Bcl6 and facilitates ASC differentiation (*Nutt et al., 2015*). Bcl6 protein levels can be regulated at both the transcriptional and post-transcriptional levels (*Duan et al., 2012*; *Saito et al., 2006*; *Takahashi et al., 2012*).

CD40 signaling plays a pivotal role in the generation of GCs (*Foy et al., 1996*). CD40 signals via the non-canonical NFκB pathway that uniquely invokes the nuclear translocation of heterodimeric RelB/p52 (*Fusco et al., 2008*; *Hömig-Hölzel et al., 2008*; *Vallabhapurapu and Karin, 2009*). Although CD40 signaling is clearly a necessary component of GC formation, a transcriptional program consistent with persistent CD40 signaling is not seen in GC B cells (*Basso et al., 2004*). Moreover, chronic CD40 signaling is known to antagonize GC B cell formation and instead promotes plasma cell differentiation (*Allman et al., 1996*; *Bolduc et al., 2010*; *Kishi et al., 2010*; *Nutt et al., 2015*; *Saito et al., 2007*). This is in part mediated by IRF4, a transcription factor promoted by RelB. However, IRF4 appears to be essential for the differentiation of both GC B cells and ASCs (*Klein et al., 2006*; *Ochiai et al., 2013*; *Sciammas et al., 2006*; *Willis et al., 2014*). Distinct DNA binding modes of IRF4 dependent on protein concentration may account for these opposing roles of IRF4 (*Ochiai et al., 2013*). To further confound things, a key cytokine secreted by $T_{fh}$ cells, IL-21, has also been shown to be an important regulator of both GC B cell and plasma cell differentiation in a B-cell intrinsic manner (*Ozaki et al., 2004*, *Ozaki et al., 2002*; *Zotos et al., 2010*).

Based on those paradoxical facts, we questioned whether GC differentiation involved multiple stages that were distinctly regulated. We hypothesized that GC B cell differentiation is completed by a stage that was not reliant on T cell-derived CD40. We observed an early peri-follicular presence of GC precursors that co-express RelB, IRF4 and BCL6, factors known to be mutually antagonistic. Using both in vivo and in vitro models, manipulating the extent or duration of T cell help and/or CD40 signaling during this early initiation stage significantly impacted the amount of precursor expansion and the capacity to transition to Bcl6$^{hi}$ GC B cells. Here we show that prolonging the availability of T cell help or CD40 agonism in vivo alters the fate of the immediate precursors away from GC path prior to follicular re-entry, whereas their removal encourages it. Thus, CD40 signaling is required for the first increase in BCL6 protein, but must cease at the immediately subsequent stage in order to allow for GC B cell progression. Based on these results, we propose a model wherein initial GC B cell differentiation is a multi-staged process, the final steps of which are shaped by a transient diminution of T cell help within its microenvironment.

## Results

### Intermediate levels of BCL6 are found in a subset of RelB$^+$ IRF4$^+$ Ag-specific B cells prior to the emergence of follicular BCL6$^{hi}$ GC B cells that lack RelB and IRF4

To examine the cellular and molecular process governing GC formation, we assessed the CD40 signaling status of Ag-specific B cells destined for the germinal center B cell fate. CD40 signals via the non-canonical NFκB pathway invoke transcriptional upregulation (*Bren et al., 2001*; *Neumann et al., 1996*), protein stabilization and nuclear translocation of heterodimeric RelB/p52 (*Fusco et al., 2008*; *Hömig-Hölzel et al., 2008*; *Vallabhapurapu and Karin, 2009*). Throughout the study, we utilized an adoptive transfer system of nitrophenyl hapten (NP)-specific B cell to investigate early B cell responses in draining lymph nodes (LNs) following footpad (*f.p.*) immunization of NP-CGG in CFA (*Figure 1A*). As expected, nuclear RelB was observed in a subset of transferred NP-specific B cells by IF histology day 2 and 3 p.i. (p.i.) (*Figure 1B*, *Figure 1—figure supplements 1* and *3*). RelB was abundant in other cells, presumably myeloid derived cells, located in T zone, IF zone and near the subcapsular sinus (SCS), but absent in follicular naïve B cells. As blocking CD40 interactions resulted in the loss of RelB expression in responding B cells, we found elevated RelB to

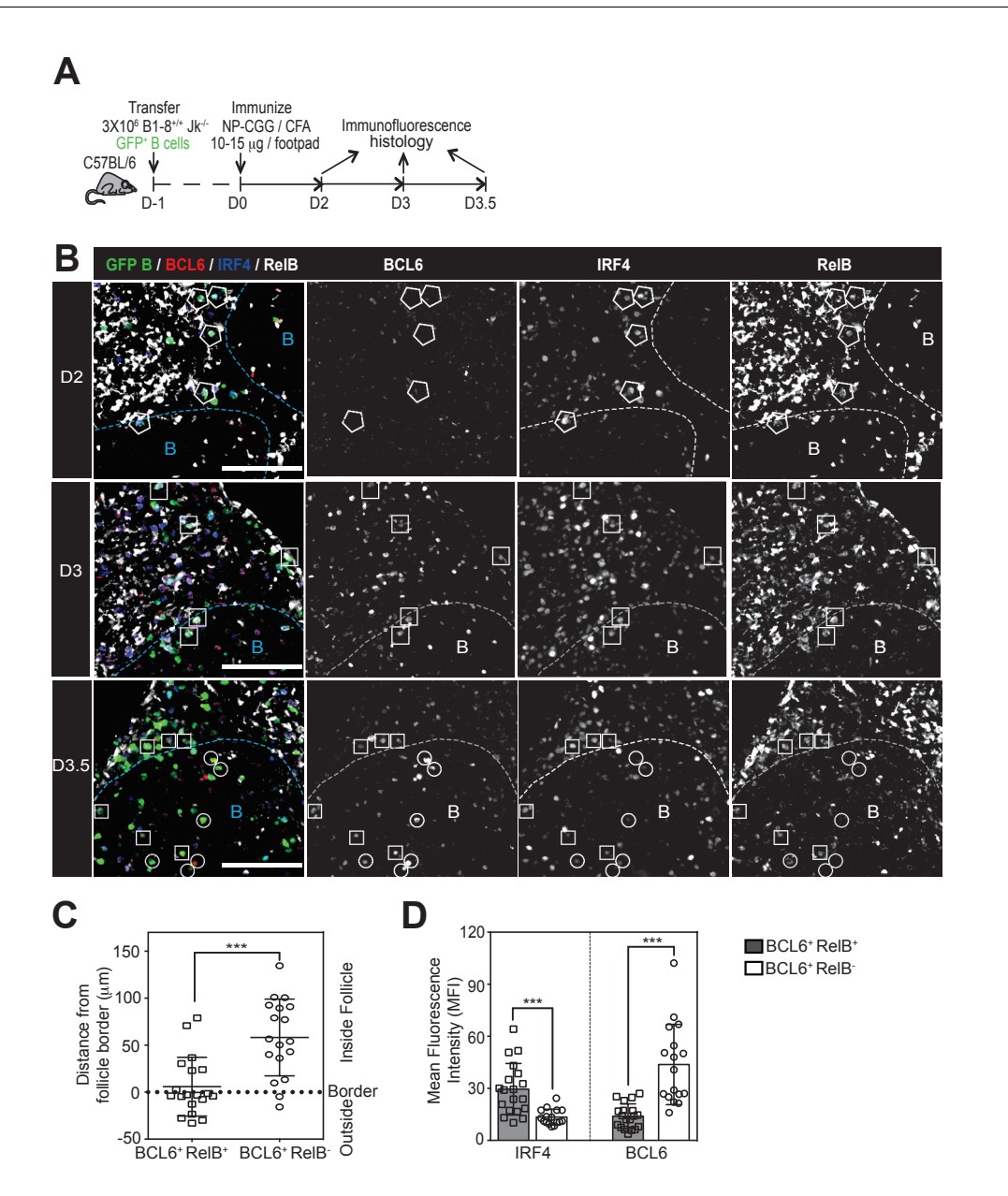

**Figure 1.** GC precursors co-express counter-regulatory factors BCL6, RelB and IRF4. (**A**) Diagram of the experimental protocol. The GFP+ NP-specific B cells were transferred into non-responding C57BL recipients, which were subsequently immunized by footpad with NP-CGG in CFA. Popliteal draining LNs were harvested and processed by day 2, 3 or 3.5 post-immunization for immunofluorescence histology. Sections were stained for BCL6, RelB and IRF4 as well as GFP to identify Ag-specific B cells. (**B**) Representative overlaid images (GFP+ B: green, BCL6: red, IRF4: blue, RelB: grey) and single channel grey images for each time point are shown.B cell follicle borders were drawn based on RelB staining. Polygons indicate GFP+ cells negative of BCL6 but positive of IRF4 and RelB; rectangles for GFP+ cells positive of BCL6, IRF4 and RelB; circles for GFP+ B cells positive of BCL6 but negative of IRF4 and RelB. Scale bars represent 100 μm. (**C–D**) The day 3.5 images acquired from three individual LNs were further analyzed by Imaris for distance replacement from follicle borders (**C**) and the Mean Fluorescence Intensity (MFI) of interested cells (**D**). Each square (n = 21) or circle (n = 16) represents the individual cell analyzed, ***p<0.001 (unpaired t-test).

The following figure supplements are available for figure 1:

**Figure supplement 1.** RelB is selectively expressed in Ag-specific responding B cells in a CD40-dependent manner during early immune responses.

**Figure supplement 2.** RelB+ cells are identified as having nuclear but not cytosolic distribution of RelB.

*Figure 1 continued on next page*

*Figure 1 continued*

**Figure supplement 3.** Absence of BCL6 staining in Ag-specific B cells of mice treated with anti-CD40L.

be a reliable indicator of CD40 signaling and used it as surrogate marker for recent signaling events (*Figure 1—figure supplement 1*).

We examined the expression levels of RelB, IRF4 and BCL6 in GFP$^+$ NP-specific B cells during the early stages of GC B cell differentiation using the adoptive transfer model described above (*Figure 1A*). Two days p.i., GFP$^+$ NP-specific B cells were found predominantly in the IF zone and at the T / B border and were RelB$^+$ and IRF4$^+$, but expressed undetectable levels of BCL6 (*Figure 1B*). BCL6 expression was not observed in NP-specific B cells after CD40 blockage, corroborating the specificity of BCL6 staining (*Figure 1—figure supplement 3*). At this point in time, nearly all RelB$^+$ responding B cells expressed elevated levels of IRF4, although the reverse was not true. Consistent with our prior study, expression of BCL6 was not apparent among NP-specific B cells until d3 p.i., a point in time when they remained largely constrained to the IF zone (*Kerfoot et al., 2011*) (*Figure 1B*). Strikingly, we found that all BCL6 expressing B cells at this time point harbored nuclear RelB and IRF4, although the BCL6 expression levels of such cells was less than observed in fully differentiated GC B cells (*Figure 1B* and data not shown; discrimination of nuclear RelB from the cytoplasmic form is demonstrated in *Figure 1—figure supplement 2*). Only a half day later (d3.5), GFP$^+$ B cells expressing higher levels of BCL6 with diminished levels of RelB and IRF4 began to emerge (*Figure 1B,D*). Image analysis comparing BCL6$^+$ RelB$^+$ cells to BCL6$^+$ RelB$^-$ cells revealed that the newly formed BCL6$^{hi}$ RelB$^-$ cells were located much deeper within follicles, whereas BCL6$^{int}$ RelB$^+$ cells resided mainly outside of follicles or close to follicular borders (*Figure 1C,D*). Thus, intermediate levels of BCL6 are first observed in RelB$^+$ B cells, suggesting that ongoing CD40 signals are important to this differentiation step.

## The BCL6$^{int}$ RelB$^+$ IRF4$^+$ population is transient and has an incomplete GC phenotype

Flow cytometry results support the conclusion that BCL6$^{int}$ RelB$^+$ IRF4$^+$ B cells temporally precede follicular BCL6$^{hi}$ GC B cells (*Figure 2A*). Consistent with the histology data, the expression of RelB in BCL6$^{int}$ IRF4$^+$ cells is significantly higher in BCL6$^{hi}$ IRF4$^{lo}$ GC B cells (*Figure 2B*). The BCL6$^{int}$ population evidenced an early and transient pattern: it emerged by 3 days pi., before the appearance of intrafollicular GC B cells, peaked at day 3.5 and rapidly declined by day 8 when GC B cells were abundant (*Figure 2C,D*). The BCL6$^{int}$ RelB$^+$ IRF4$^+$ nascent GC B cell precursors displayed a partial GC phenotype (*Figure 2E*). They expressed lower levels of PNA binding and Fas and less repression of the BCL6 target gene CD38 compared to their BCL6$^{hi}$ GC B cell counterparts (*Figure 2E*). Interestingly, significantly higher levels of CD86 were observed among the BCL6$^{int}$ RelB$^+$ IRF4$^+$ GC precursors. It is important to note that these markers are not exclusive to GCs during the early stages of the response, and that other activated B cell subsets not expressing BCL6 can also show elevated levels of Fas and PNA binding (*Figure 3*). Together these results implicate BCL6$^{int}$ RelB$^+$ IRF4$^+$ B cells as a GC precursor population that immediately precedes BCL6$^{hi}$ RelB$^{lo}$ IRF4$^{lo}$ GC B cells. From here on, we refer to BCL6$^{int}$ RelB$^+$ IRF4$^+$ B cells as the GC precursors (or pre-GC) and BCL6$^{hi}$ RelB$^{lo}$ IRF4$^{lo}$ cells as GC B cells.

## The coordinated transition of GC precursors to BCL6$^{hi}$ GC B cells is associated with cell division and a loss of RelB

Our data suggest that GC formation requires a transition from a BCL6$^{int}$ GC precursor evidencing recent CD40 signaling (RelB+), to a BCL6$^{hi}$ GC B cell lacking evidence of CD40 signaling (RelB-), despite continued expression of surface CD40 at comparable levels (data not shown). To assess whether a transition from GC precursor to GC B cell phenotype correlated with cell division in vivo, CFSE-labeled NP-specific B cells were transferred into congenic mice prior to immunization (*Figure 4A*). We found that the majority of antigen specific B cells underwent six rounds of division 3 days p.i., and two or more additional divisions over the following day (*Figure 4B,C*).

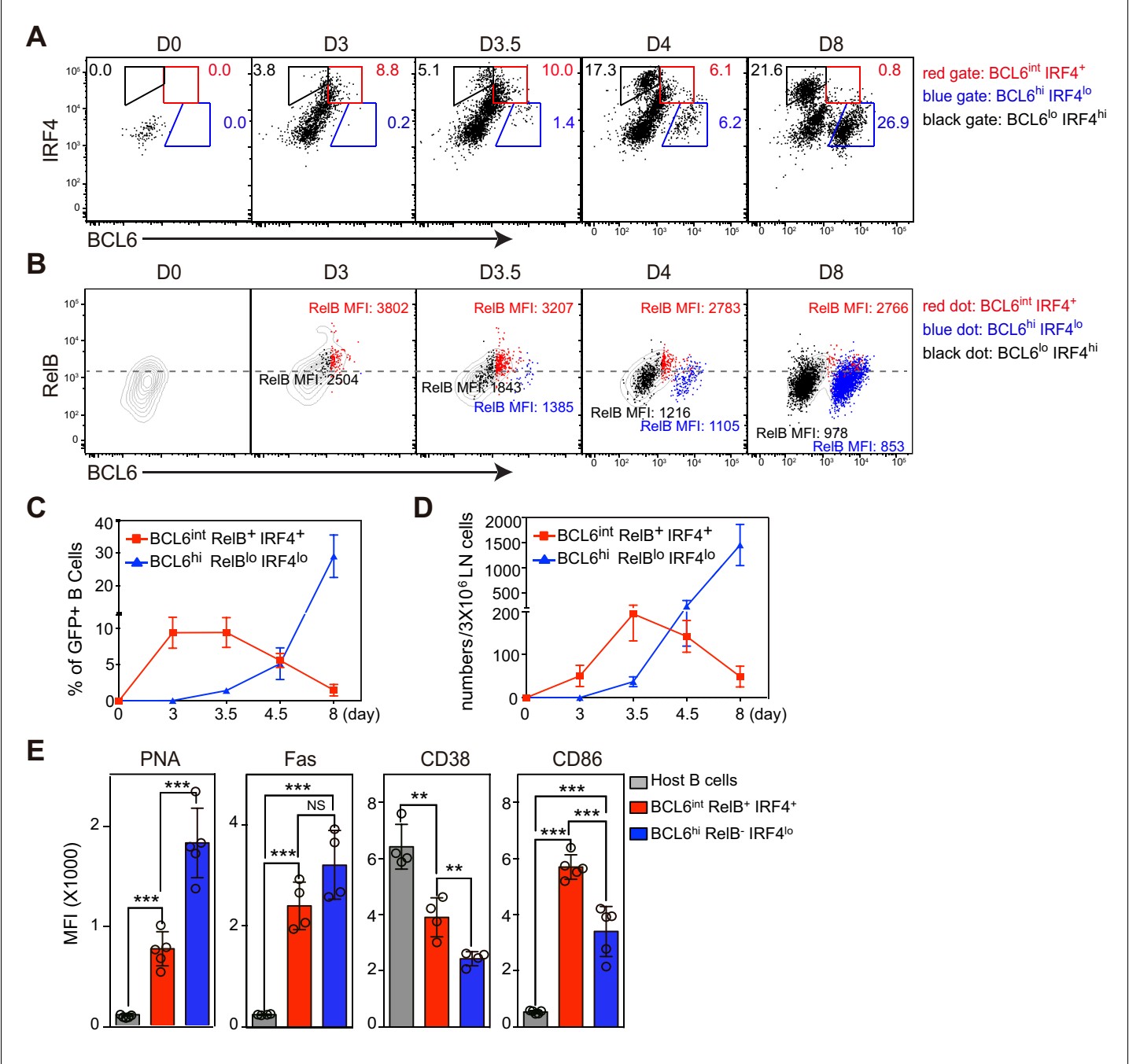

**Figure 2.** BCL6int RelB+ IRF4+ Ag-specific B cells emerge early during immune responses and have partial GC phenotypes. (**A–E**) Flow cytometric analysis of draining LN Ag-specific B cells. GFP+ NP-specific B cells were transferred into C57BL/6 recipients, which were subsequently immunized by footpad with NP-CGG in CFA. Popliteal draining LNs were harvested and stained 0, 3, 3.5, 4.5 and 8 days *p.i.* Data are representative of three independent experiments. (**A**) Expression of BCL6 and IRF4 in transferred NP-specific B cells from each time point. BCL6int IRF4+, BCL6hi IRF4lo and BCL6lo IRF4hi cells were gated as indicated. (**B**) Representative contour plots of RelB and BCL6 expression. Gated BCL6int IRF4+ (red dots), BCL6hi IRF4lo (blue dots) and BCL6lo IRF4hi (black dots) cells were backgated and overlaid onto total NP-specific B cell contours. The MFI of RelB in each population was indicated with representative color. (**C–D**) Shown are average percentage ± SEM of BCL6int RelB+ IRF4+ cells and BCL6hi RelB- IRF4lo cells among transferred B cells (**C**) and their average numbers ± SEM per $3 \times 10^6$ LN cells (**D**) (n = 4 at each time point from one experiment). (**E**) Bar graphs show the average MFI ±SEM of PNA, Fas, CD38, CD86 and CD23 in indicated cell subsets 3.5 days after NP-CGG immunization. NS no significant difference, **p<0.01, ***p<0.001 are compared as indicated (one-way ANOVA).

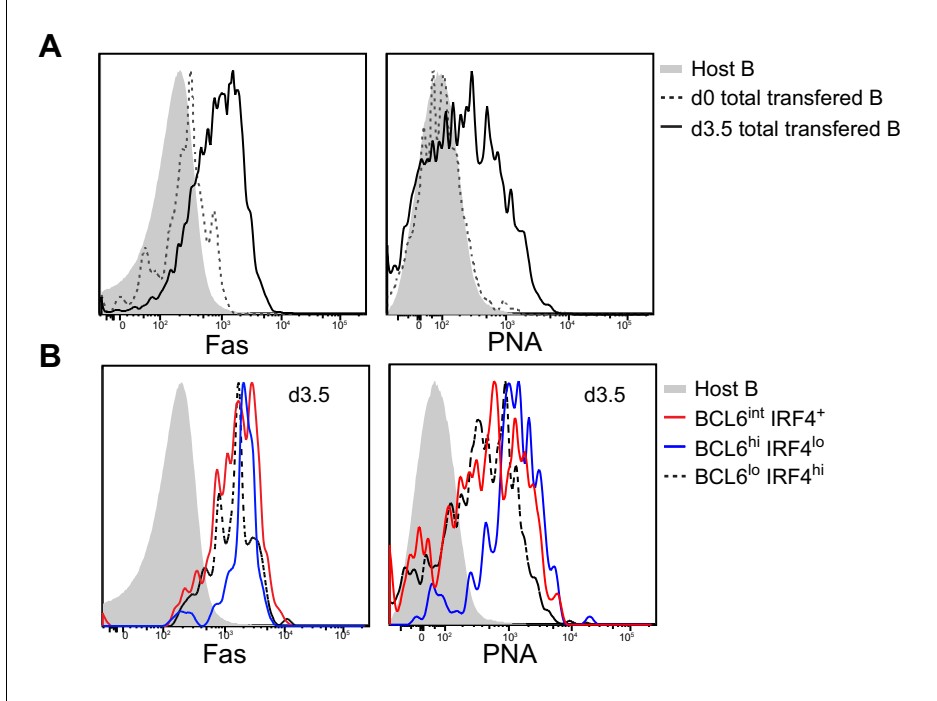

**Figure 3.** Multiple activated B cell subsets have elevated Fas and PNA-binding levels, including BCL6int IRF4+ cells. GFP+ NP-specific B cells were transferred into C57BL/6 recipients, followed by *f.p.* immunization of NP-CGG in CFA. Popliteal draining LNs were stained for flow cytometric analysis. (A–B) Representative histograms of Fas experssion and PNA-binding by transfered B cells (total or indicated subsets) and control populations (host non-responding B cells) in draining LNs.

With these immunization conditions, most GC precursors were found to have undergone 4–6 divisions (*Figure 4B,C*). In contrast, GC precursors were significantly less frequent among cells that have undergone seven or more divisions, coincident with the emergence of GC B cells at divisions 6–7 (*Figure 4B,C*). IRF4hi cells lacking Bcl6, presumably early plasmablasts, emerge with slightly faster kinetics than GC B cell and continue to undergo additional cell divisions (*Figure 4—figure supplement 1*).

The lack of further divisions of GC precursors could be explained by either cell cycle arrest, exacerbated cell death or further differentiation to GC B cells. Cell cycle analysis indicated that GC precursors were predominantly in cell cycle and not evidently apoptotic compared to non-responding B cells (*Figure 4—figure supplement 2A,B*). Therefore, the lack of GC precursors that have undergone more than six rounds of divisions does not appear to reflect a failure to enter into cell cycle or an increase of apoptotic cell death, but may result from differentiation into GC B cells. We further questioned whether expression levels of differentiation fate-determining transcription factors are linked to cell division of GC immediate precursors. In contrast to relatively stable expression of BCL6 and IRF4, a gradual reduction of RelB through rounds of cell divisions was observed (*Figure 4D*). Taken together, the results suggest the coordinated transition of GC precursors to GC B cells is associated with cell division and a loss of RelB expression.

## A smaller immunogen dose increases the frequency of GC B cell differentiation over fewer rounds of cell division

We hypothesized that the reduction of RelB levels in GC precursors at later divisions (*Figure 4D*) reflected a lack of Th cell engagement. It has been suggested previously that the magnitude of Ag presentation by responding B cells decreases over the course of sequential cell divisions (*Thaunat et al., 2012*). Immunogen dose might therefore influence how many rounds of cell division are necessary for GC B cell formation.

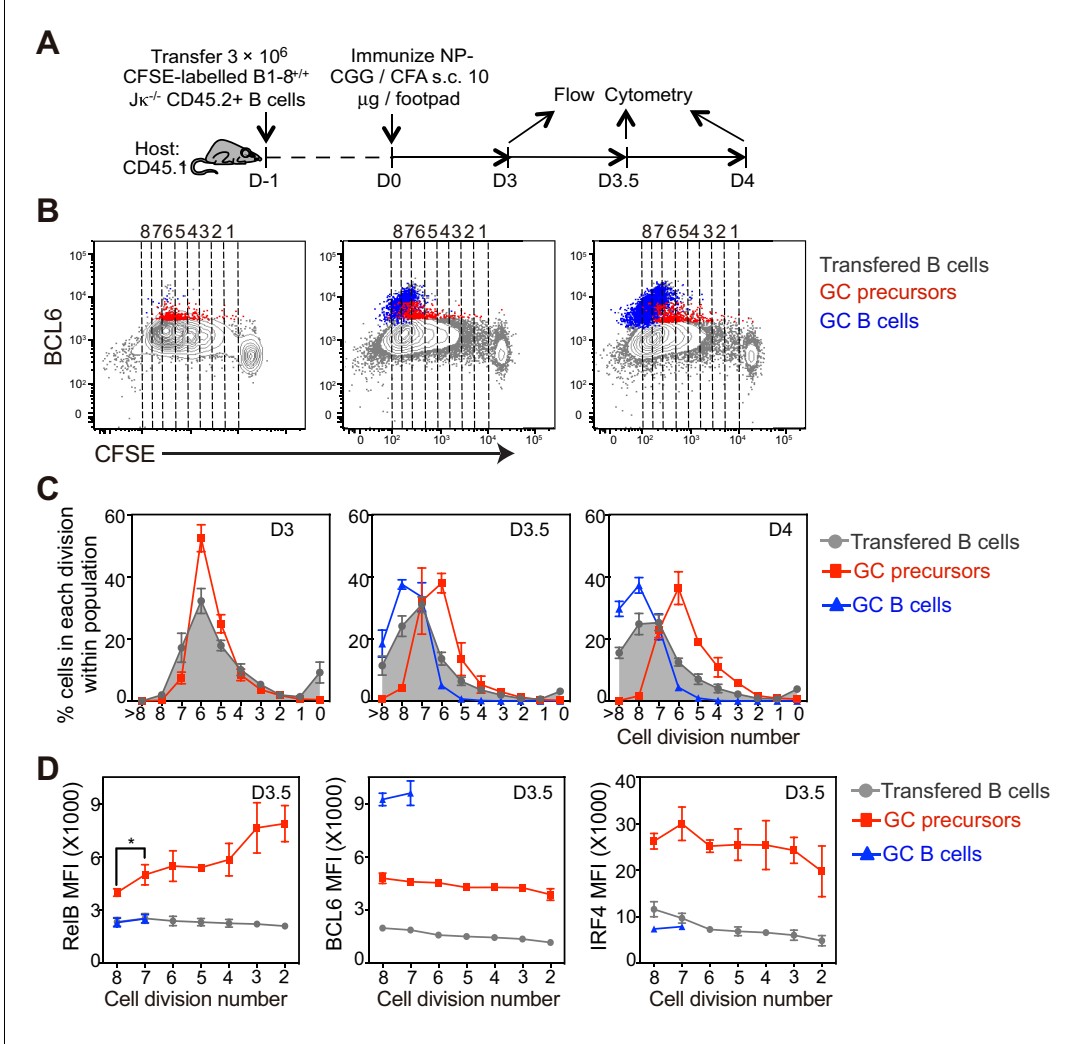

**Figure 4.** The coordinated transition of GC precursor to BCL6hi GC B cells is associated with cell division. (**A**) Diagram of the experimental protocol. CFSE-labeled NP-specific B cells were transferred into CD45.1+ congenic recipients, followed by f.p. immunization with NP-CGG in CFA. Popliteal draining LNs were harvested and stained for flow cytometry analysis 3, 3.5 and 4 days *p.i.* (**B**) Representative dot plots of three independent experiments show BCL6 expression relative to CFSE dilution in total transferred B cells (grey dots), GC precursors (red dots) and GC B cells (blue dots). Cell divisions (lines and numbers) up to 8 were assigned based on CFSE serial dilutions of total transferred B cells. (**C**) Graphed average percentage ± SEM of cells at each number of division within population (n = 4 from per experiment). (**D**) By day 3.5 *p.i.*, indicated cell subsets were further analyzed for the average MFI ±SEM of BCL6, RelB and IRF4, plotted as a function of cell division. *p<0.05 (repeated measures ANOVA).

The following figure supplements are available for figure 4:

**Figure supplement 1.** BCL6lo IRF4hi antigen-specific B cell emerge and expand through cell divisions with kinetics that differ from GC precursors or GC B cells.

**Figure supplement 2.** GC precursors are in active cell cycle and not apoptotic.

To examine the impact of hapten dose, recipients of CFSE labeled B cells were immunized with either 15 ug NP-CGG or 1.5 ug of NP-CGG supplemented with 13.5 ug CGG (*Figure 5A*). In order to provide comparable conditions for DC antigen presentation and early Th cell development, the amount of the carrier protein CGG did not vary. Regardless of the dose of haptenated protein, the extent of proliferation of transferred B cells was similar 3.5 days p.i. (*Figure 5C*). A low dose of NP-CGG resulted in a smaller percentage of GC precursors but a significantly larger percentage of GC B cells (*Figure 5B,C*). Moreover, the quantity of Ag affects the frequency of differentiation

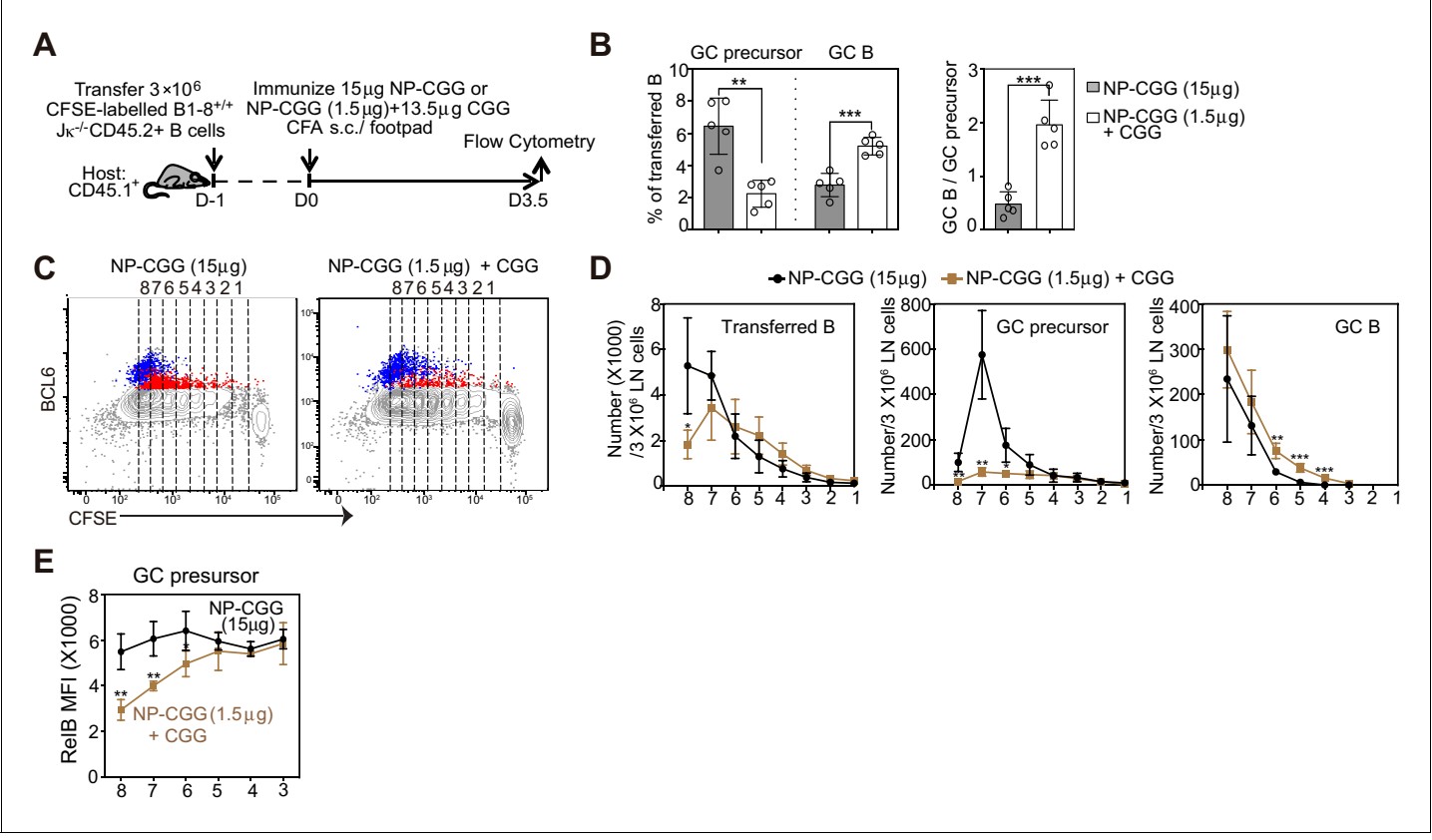

**Figure 5.** A smaller immunogen dose promotes the transition of precursors to GC B cells. (**A**) Diagram of the experimental protocol. CFSE-labeled NP-specific B cells were transferred into CD45.1+ congenic recipients, followed by f.p. immunization of NP-CGG of various doses (either 1.5 μg NP-CGG +13.5 μg CGG or 15 μg NP-CGG) in CFA. Popliteal draining LNs were harvested and stained for flow cytometric analysis 3.5 days *p.i.*. (**B**) Percentage of GC precursors or GC B cells among transferred B cells (left panel), and the ratio of GC B cells to GC precursors (right panel). (**C**) Representative dot plots of BCL6 expression of transferred B (grey), GC precursors (red) and GC B cells (blue) relative to CFSE dilution. Cell divisions (lines and numbers) up to eight were assigned based on CFSE serial dilutions of transferred B cells. (**D**) Number of CD45.2+ B cells, GC precursors or GC B cells at each division per $3 \times 10^6$ LN cells. (**E**) Average MFI ±SEM of RelB in GC precursors at each division. Data are representative of 2 independent experiments with n = 5 in each group. *p<0.05, **p<0.01, ***p<0.001 (unpaired t-test (a) or multiple t-test (d,e)) compared as indicated or between different dosage groups at respective cell division number.

relative to rounds of cell division (*Figure 5C,D*). With the lower dose of NP-CGG, GC B cells emerged at divisions 4–6, suggesting that GC precursors are predisposed to differentiate with fewer rounds of divisions when mice received lower doses of Ag (*Figure 5C,D*). With a lower Ag dose, a steep reduction in the RelB MFI observed at later precursor cell divisions was associated with an accelerated rate of transition to the matured GC B cell state (*Figure 5E*). Collectively, these results indicate that the quantity of Ag at immunization impacts the kinetics and number of divisions required for the reduction of RelB in GC precursors and their transition to GC B cells.

## Sustained T cell help at the initiation stage diverts the fate of GC precursors towards plasmablast formation

To determine whether persistent T cell help discourages the differentiation of GC B cells, we varied the extent of T-cell help in mice 3 days after the initial immunization, a critical point in time for the GC B cell transition in vivo. As illustrated in *Figure 6A*, mice were either additionally injected with NP-CGG at day 3 to simulate a persistent antigen presence and maintain presentation by NP-specific B cells, or with agonistic anti-CD40 to mimic the delivery of this component of T-cell help. Interestingly, the introduction of either NP-CGG or anti-CD40 at day 3 markedly impeded the GC B cell transition but significantly increased the frequency of **their** immediate precursors (*Figure 6B,C*).

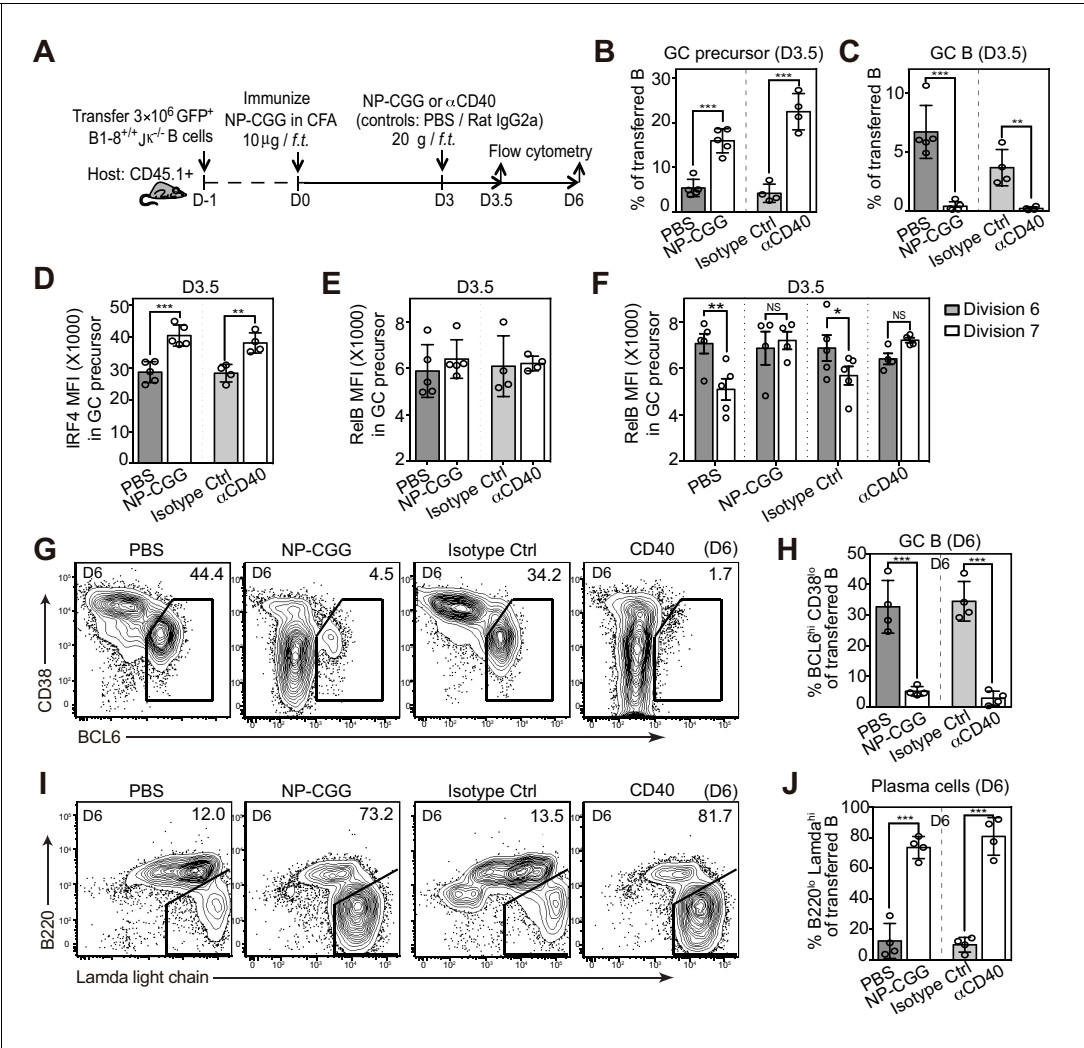

**Figure 6.** Potentiating T cell help at the initiation stage discourages the transition of GC precursors to GC B cells, and diverts their fate towards plasma cells. (A) Diagram of the experimental protocol. 3 days after the initial immunization, recipient mice received an additional injection of either soluble NP-CGG, agonistic anti-CD40 (clone: FGK), PBS or isotype control (Ctrl). Draining LN cells were harvested and stained for flow cytometric analysis by 3.5 and 6 days *p.i.* (B–F) Percent GC precursors and GC B cells among transferred B cells 3.5 days p.i. (B and C); The average MFI ±SEM of IRF4 and RelB in GC precursors (D and E); and the average MFI of RelB expression in GC precursors at division 6 and 7 defined by CFSE serial dilutions of the transferred B cells (F). (G–J) Representative contour plots and bar graph showing the expression of BCL6 and CD38 (G) and percentage of GC B cells (BCL6hi CD38lo) among transferred B cells 6 days p.i. (H); the expression of Ig Lamda light chain and B220 (I) and percentage of plasma cells (Lamdahi B220lo) (J). Data are representative of at least two independent experiments with n = 4–5 in each group. NS no significant, *p<0.05, **p<0.01, ***p<0.001 (unpaired t-test (b,c,d,e,h.j) or paired t-test (f)).

Although overall RelB levels were not affected, IRF4 levels were elevated in GC precursors compared to treatment controls (*Figure 6D,E*). Further examination of CFSE-labeled B cells revealed that GC precursors did not display the typical pattern of RelB reduction at later cell divisions when either additional antigen or agonistic anti-CD40 is introduced (*Figure 6F*). These results are consistent with the idea that prolonged Ag exposure and T-cell help during the GC transition phase can drive the formation of GC precursors and impedes their transition to GC B cells. This did not represent just a delay in the kinetics, as the decrease in the GC population was evident 6 days p.i. as well (*Figure 6G,H*). In sharp contrast, both treatments strongly promoted the formation of plasmablasts (*Figure 6I,J*). Together the results suggest that prolonged T-cell help or CD40 agonism during this critical transitional phase skews the fate of GC precursors toward plasmablast differentiation and

away from GC B cell formation. Although the results support this conclusion, alternative explanations for the impact of larger immunogen doses cannot be excluded.

## A reduction in T cell help differentially impacts GC precursors and GC B cells

We questioned whether depletion of CD4+ T cells or CD40L blockade would facilitate the transition to GC B cells from existing GC precursors (*Figure 7A*). Although both treatments resulted in a rapid reduction in detectable GC precursors just a half day later, GC B cell frequencies remained comparable (*Figure 7B–D*). This short-term blockade of CD40L did not lead to an increase in apoptosis of GC precursors (*Figure 7E–F*) or cell cycle arrest (data not shown). Therefore transient disruption of B and T interactions or CD40 signaling in vivo does not prevent the furthered differentiation of precursor cells, and instead promotes this transition at this early phase of the immune response. Prolonged exposure to CD40L blockade, as expected, prevented GC B cell expansion subsequently (*Figure 7—figure supplement 1*). To further explore whether a transient disengagement from cognate T cells facilitates this transition in vivo, we examined the tendency of GC precursors and BCL6$^{hi}$ GC B cells to form cognate T/B pairs 3.5 days p.i. Whereas the majority of RelB$^+$ BCL6$^+$ B cells (64%, n = 86 from tissue sections of 5 mice) were in direct contact with OVA-specific T cells, few RelB$^{lo}$ BCL6$^+$ B cells were in contact with cognate T cells (17%, n = 79) (*Figure 7G,H*). Interestingly, RelB$^{lo}$ BCL6$^+$ B cells could still be found in the IF zone amidst abundant cognate T cells (*Figure 7G*). Thus a disengagement from T cell help is associated with reduced RelB levels in GC precursors and promotion of GC B cell differentiation.

## Deprivation of T cell derived stimuli promotes the in-vitro generation of Bcl6$^{hi}$ IRF4$^-$ RelB$^-$ B cells

Our in vivo studies demonstrated that RelB expression, an indicator of CD40 signaling, in GC precursors is attenuated before BCL6 levels are increased. We established an in vitro culture system that generates BCL6$^{int}$ RelB$^+$ IRF4$^+$ GC precursors prior to BCL6$^{hi}$ RelB$^-$ IRF4$^-$ GC B cells, recapitulating the in vivo observations. As illustrated in *Figure 8A*, freshly isolated primary B cells were pulse-stimulated with anti-IgM to mimic in vivo antigen recognition, followed by seeding onto BAFF-expressing FL-YB cells for survival support (*Magari et al., 2011*). At day 3 of culture, T cell associated stimuli (αCD40 and IL-4/21) were washed away and cells were re-cultured in medium only for an additional half-day before analysis. BCL6 expression was substantially increased in B cells in cell cultures provided with continuous stimulation of αCD40 and IL-4/21, however, BCL6$^+$ cells remained RelB$^+$ and IRF4$^+$, resembling GC precursors (*Figure 8B,C*). After a half-day deprivation of αCD40 and IL-4/21, a population of cells with substantially higher expression of BCL6 was generated (*Figure 8B*). These BCL6$^{hi}$ cells expressed low levels of RelB and IRF4, resembling GC B cells (*Figure 8C*). Moreover, fewer BCL6$^{int}$ RelB$^+$ IRF4$^+$ B cells were observed after discontinuation of αCD40 and IL-4/21 (*Figure 8C*). Notably, removal of either αCD40 or IL-4/21 alone was insufficient to allow robust generation of the GC-like population (data not shown).

We further analyzed the phenotype of BCL6$^{int}$ RelB$^+$ IRF4$^+$ or BCL6$^{hi}$ RelB$^{lo}$ IRF4$^{lo}$ cells generated in αCD40 and IL-4/21 washout cultures. BCL6$^+$ RelB$^+$ IRF4$^+$ had a partial suppression of CD38 and elevated CD86 levels despite high levels of GL7 and Fas, similar to their in-vivo generated GC precursor counterparts (*Figure 8D*). Compared to the pre-GC-like population, the BCL6$^{hi}$ RelB$^{lo}$ IRF4$^{lo}$ cells show reduced CD38 and CD86 expression (*Figure 8D*). In addition, cell division analysis of washout cultures indicated that BCL6$^{hi}$ RelB$^{lo}$ cells underwent several more rounds of cell division than BCL6$^{int}$ RelB$^+$ cells (*Figure 8E*), suggesting that BCL6$^{hi}$ cells differentiated from BCL6$^{int}$ RelB$^+$ precursors that have undergone at least one further division without T cell-associated stimuli (*Figure 8E*).

To determine whether transcriptional and / or posttranscriptional regulation accounts for the shifts in Bcl6 protein levels between the BCL6$^{int}$ and BCL6$^{hi}$ populations, we utilized a PrimeFlow RNA assay (*Allison et al., 2016*; *Porichis et al., 2014*; *Soh et al., 2016*) that allows for the detection of *bcl6* mRNA in combination with intracellular staining of protein BCL6, RelB and IRF4 at single cell level. Hybridization and amplification of *bcl6* binding probes indicate that both transcriptional and post-transcriptional regulation are involved in these phases of differentiation. Compared to αIgM-pulsed B cells with or without αCD40 activation, B cells with a continuous exposure to αCD40 and

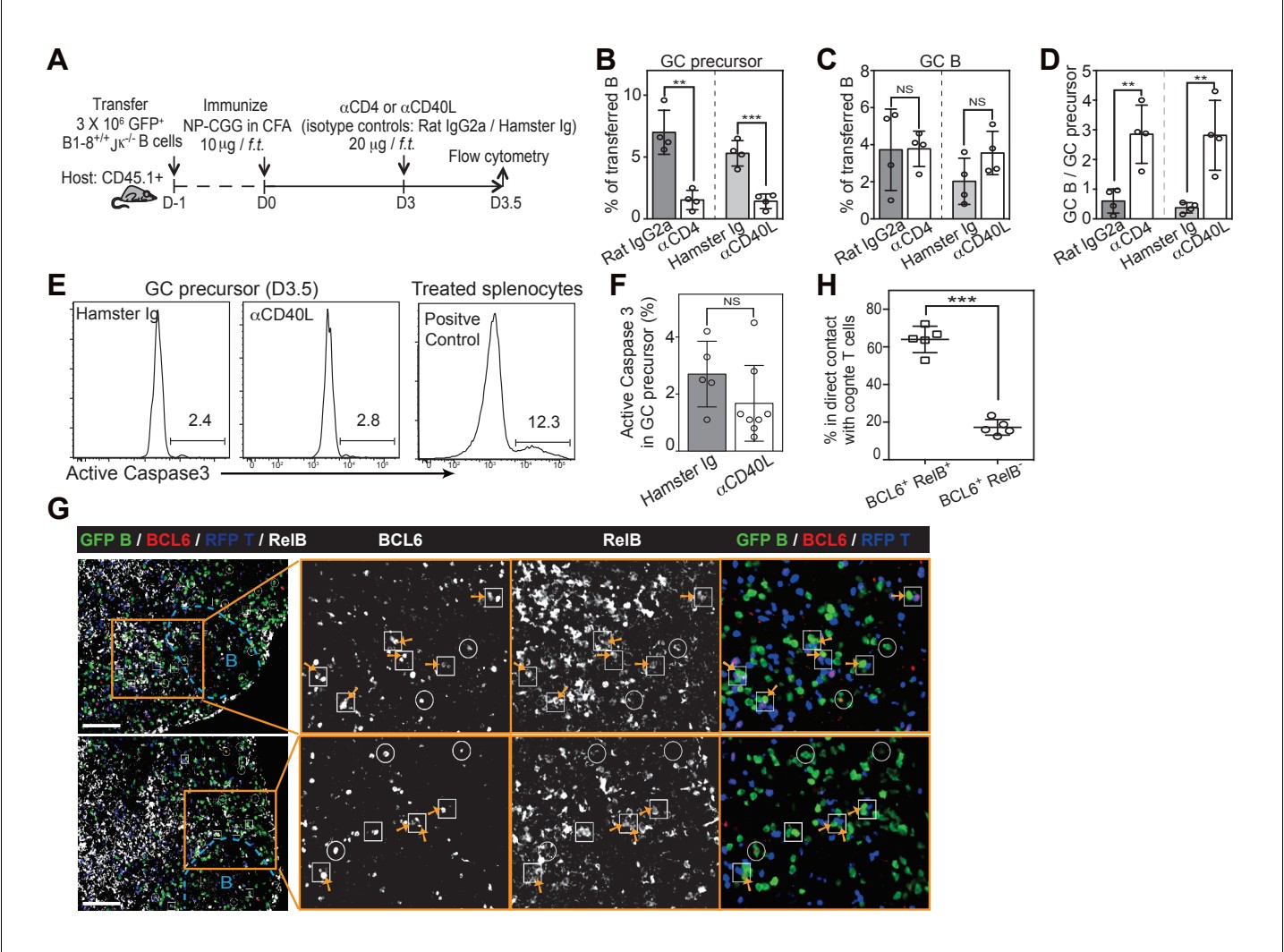

**Figure 7.** Disruption of T/B interaction differentially impacts GC precursors and GC B cells. (A–F) Diagram of the experimental protocol (A). By day 3 post-immunization, 20 μg αCD4 or αCD40L was further injected locally to delete CD4 T cells or block CD40 ligation on B cells. Popliteal draining LNs were harvested and stained for flow cytometry analysis by day 3.5 post-immunization. Two independent experiments were carried with n = 4–5 mice in each group. (B–C) Bar graphs showing the percent GC precursors (B) and GC B cells (C) in transferred B cells. (D) The rate of transition is indicated by the ratio of GC precursors over GC B cells. (E) Representative plots showing apoptosis (active caspase 3+) in gated GC precursors after CD40L blockage; 2 hr Staurosporine-treated murine splenocytes were used as positive control to verify active caspase 3 staining. (F) Bar graph showing no significant difference in the induction of Active Caspase 3 within GC precursors upon acute blockage of CD40L. (G–H) NP-specific GFP+ B cells and OVA-specific RFP+ T cells were transferred into C57BL/6 recipients, followed by immunization of NP-OVA in CFA. 3 days and 6 hr later, popliteal draining LNs were harvested and stained for histology. (G) Sections were stained for BCL6 and RelB as well as GFP and RFP to identify Ag-specific B and T cells. BCL6+ RelB+ (square) or BCL6+ RelB- (circle) GFP+ B cells were indentified, and those in the close contact with cognate T cells were indicated with yellow arrows. Scale bars represent 100 μm. (H) Total BCL6+ RelB+ (n = 86) and BCL6+ RelB- (n = 79) from sections of 5 different LNs were revealed and graphed for % in direct contact with cognate T cells. NS no significant difference, **p<0.01, ***p<0.001 (unpaired t-test).

The following figure supplement is available for figure 7:

**Figure supplement 1.** In contrast to the initial transition of pre-GC to BCL6hi GC B cells, T cell help is required for the furthered maintenance or expansion of GC B cells.

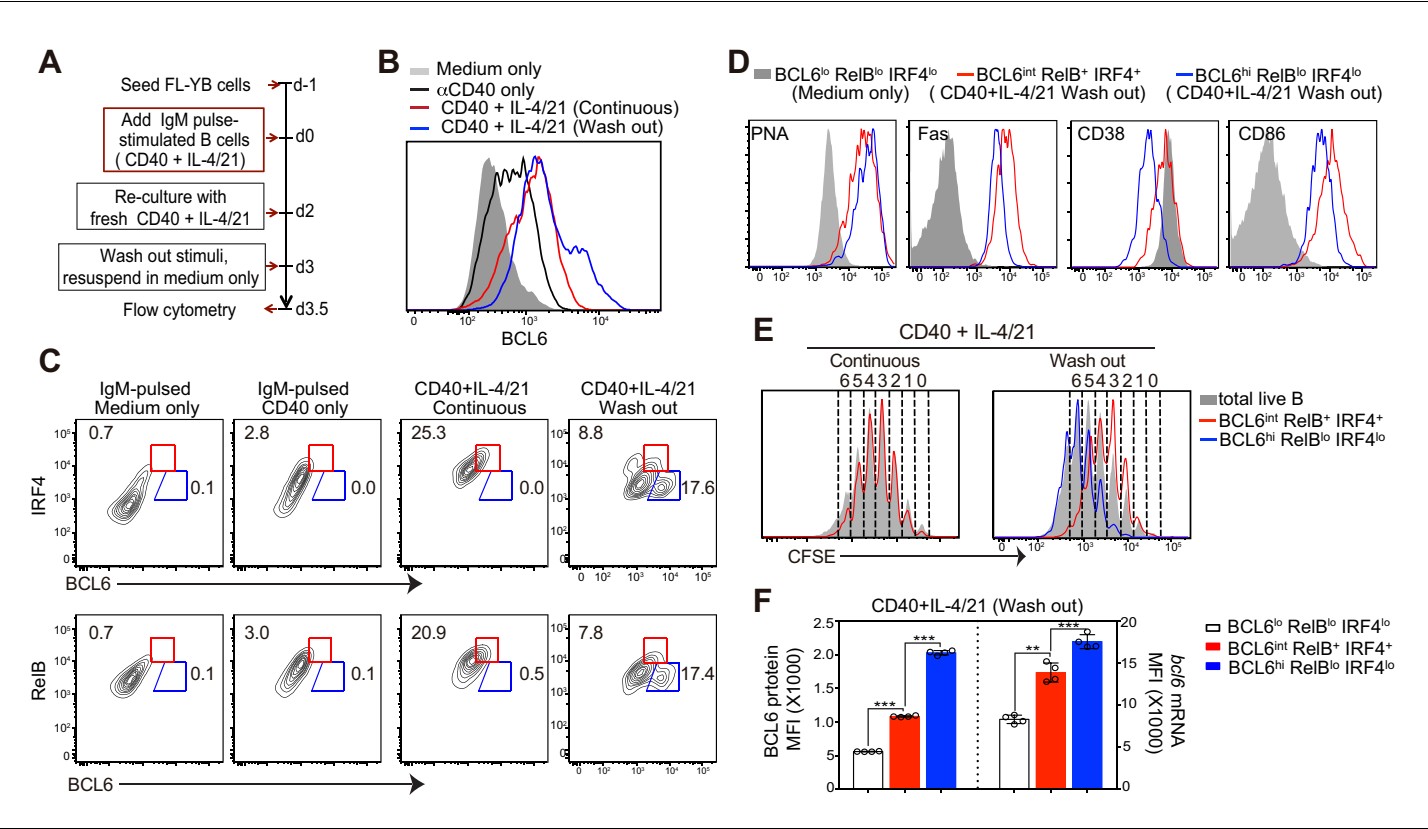

**Figure 8.** Deprivation of T-cell derived stimuli is required for the maturation of BCL6hi GC B cells in vitro. (**A**) Schematic of the culture system using primary murine B cells. (**B**) Overlay of BCL6 expression in live B cells under different culture conditions as indicated. (**C**) Contour plots of BCL6 expression by live B cells relative to either RelB or IRF4 in different culture conditions. Red gates indicate BCL6int RelB+ / IRF4+, blue gates BCL6hi RelBlo / IRF4lo. (**D**) Histogram overlay of the expression of GC phenotypic markers. Cells were gated as indicated from wash-out cultures that had medium with anti-CD40 + IL-4/21 removed after 3 days of culture. The wash-out populations are compared to BCL6lo RelBlo IRF4lo cells from cultures with medium only (solid grey). (**E**) Histogram overlay of CFSE dilutions of BCL6int RelB+ / IRF4+ (red), BCL6hi RelBlo / IRF4lo (blue) and total live B cells (grey) gated from the indicated culture conditions. Cell division was manually gated based on CFSE dilution of total live B cells. (**F**) By using Primeflow assay to detect bcl6 mRNA thorough flow cytometry, gated BCL6lo RelBlo IRF4lo, BCL6int RelB+ IRF4+, and BCL6hi RelBlo IRF4lo cells were further analyzed for the MFI of BCL6 protein and mRNA. **p<0.01, ***p<0.001 (paired t-test). Data shown above are representative of at least three independent experiments.

The following figure supplements are available for figure 8:

**Figure supplement 1.** Flow cytometric analysis of BCL6 protein and bcl6 mRNA.

**Figure supplement 2.** BCL6int precursors, not BCL6lo cells, give rise to BCL6hi GC B cells in vitro.

IL-4/21 had higher levels of both Bcl6 mRNA and protein (*Figure 8—figure supplement 1A,C*). Washout of αCD40 and IL-4/21 further enhanced the transcription of *bcl6* mRNA in cultured B cells, which mostly correlated with protein levels (*Figure 8—figure supplement 1A,B and C*). Consistent with this, GC B cell precursors (BCL6int RelB+ IRF4+) expressed much higher levels of *bcl6* mRNA than that of BCL6lo RelBlo IRF4lo cells of washout cultures (*Figure 8F*). Higher BCL6 protein levels in GC B cells (BCL6hi RelBlo IRF4lo) cells was associated with yet another increase of *bcl6* mRNA (*Figure 8F*). Notably however, BCL6hi GC B cells evidenced a significantly greater increase of BCL6 protein than that of *bcl6* mRNA, suggesting both transcriptional and post-transcriptional regulation of BCL6 expression occurs during the transition to the BCL6hi GC B cell state (*Figure 8F*).

To directly evaluate which population gives rise to BCL6hi RelBlo IRF4lo cells, sorted cells were re-cultured for differentiation. To unambiguously identify Bcl6 expressing cells based on their level of expression, we employed a strain of mice that express a Bcl6-YFP fusion protein under the direction

of the Bcl6 promoter (*Baumjohann et al., 2011*; *Kitano et al., 2011*). Three day cultures of *Bcl6*$^{yfp/+}$ splenocytes with continuous stimulation of αCD40 and IL-4/21 demonstrated that YFP expression mirrored that of Bcl6 protein with high fidelity (*Figure 8—figure supplement 2A*). In order to enrich for viable cells distinguishable by their level of Bcl6 levels, we sorted YFP$^{hi}$ CD38$^{hi}$ CD86hi and YFP$^{lo}$ CD38$^{hi}$ CD86$^{hi}$ from cultured cells. Due to the time spent on cell staining and sorting, the initial culture period was shortened from 3 days to 2.5 days (*Figure 8—figure supplement 2B*). Sorted cells were re-cultured in medium only in FL-YB-seeded wells for an additional half-day before analysis (*Figure 8—figure supplement 2B*). Unsorted cells under washout conditions (positive control) were also deprived of T cell associated stimuli by day 2.5 of culture and likewise replated onto FL-YB-seeded wells. Although the shortened initial culture with αCD40 and IL-4/21 results in fewer BCL6$^{hi}$ RelB$^-$ IRF4$^-$ GC B cells, the results indicate that only the YFPhi population gives rise to BCL6$^{hi}$ IRF4$^{lo}$ RelB$^{lo}$ cells, but the YFPlo population does not (*Figure 8—figure supplement 2C and D*). The sorted YFP$^{hi}$ cells generated two-fold more BCL6hi cells than the unsorted population. Although the rate of differentiation was lower than we expected, it is likely that the long duration of cell staining and sorting process compromised their further differentiation during the subsequent 12 hr re-culture.

Taken together, the in-vitro cultures confirmed that a physical separation of activated B cells from the CD40 agonism and cytokines typically associated with Th cells encourages the generation of BCL6$^{hi}$ RelB$^{lo}$ IRF4$^{lo}$ cells expressing a GC B cell phenotype. Whereas transition of the Bcl6$^{lo}$ to the Bcl6$^{int}$ state involves primarily Bcl6 gene transcription, the transition to the Bcl6$^{hi}$ state also evidences post-transcriptional regulation of Bcl6 mRNA levels.

## Discussion

The results presented here shed light on a longstanding conundrum; how can CD40 signaling be required for B cell commitment to the GC lineage while also inhibiting its formation? We find that GC B cell differentiation is a two-stage process; (1) a T-cell contact and CD40-dependent phase that generates RelB+ IRF4+ BCL6$^{int}$ GC precursors and (2) a phase immediately following that is prevented by further CD40 signaling but nevertheless needed to complete their transition to BCL6$^{hi}$ GC B cells. The immediate precursors to GC B cells are poised at an incomplete state of differentiation that can be easily diverted by extending T cell help or by introducing anti-CD40. In this way, the impact of CD40 signaling depends on the distinct stage of GC B cell formation, promoting precursor formation, but diverting precursor progression once formed. Together our results are consistent with the idea that the transition of GC precursors to the Bcl6$^{hi}$ state is discouraged by Th-derived CD40L during the early stages of adaptive immune responses.

Our results indicate that the initial low level of BCL6 expression is limited to B cells that also harbor nuclear RelB and IRF4. A recent report indicates that the transient expression of IRF4 is also required for proper GC B cell progression (*Ochiai et al., 2013*). Notably, IRF4 is required for both ASC and GC development, although its sustained expression preferentially promotes ASC associated genes, subverting the GC pathway (*Kwon et al., 2009*; *Ochiai et al., 2013*). The requirement for prior IRF4 expression had come as a bit of a surprise since IRF4 is known to directly bind to and represses the Bcl6 promoter (*Basso et al., 2010*; *Saito et al., 2007*). Thus GC precursors co-express transcription factors known to be mutually antagonistic. In addition to transcriptional regulation of Bcl6 gene expression, post-transcriptional regulation can also modulate Bcl6 protein levels (*Duan et al., 2012*; *Saito et al., 2006*; *Takahashi et al., 2012*). Interestingly, the comparison of Bcl6 RNA levels to protein levels under varied in vitro culture conditions suggests that each stage of differentiation may differ in this regard. Whereas RNA and protein levels appeared to have a mainly linear correlation within Bcl6$^{int}$ GC precursors, there was a disproportionate increase of Bcl6 protein within Bcl6$^{hi}$ GC B cells generated in vitro, suggesting that some form of post-transcriptional regulation may be more involved in the transition to the Bcl6$^{hi}$ GC B cell state.

Within mature GCs, BCL6 controls B cell differentiation by regulating cell cycle genes and some terminal differentiation factors (*Bunting and Melnick, 2013*). Repressed BCL6 target genes in dark zone GC B cells include cd38, irf4 and prdm1, however precursor B cells displayed only a partial GC phenotype and have not yet fully suppressed surface expression of CD38. This might reflect either a requirement for higher levels of BCL6 or alternatively an inability of precursor cells to express other co-repressors necessary for effective transcriptional repression (*Basso and Dalla-Favera, 2012*;

*Béguelin et al., 2013*; *Hatzi et al., 2013*). Interestingly, GC precursors retain some plasticity in that they can still give rise to ASC during this critical transitional time period with either larger amounts of T cell help or CD40 ligation. It is tempting to speculate that the co-expression of RelB, IRF4 and BCL6 together achieve a transcriptional program compatible with some degree of pluripotency.

What might mediate the transition of RelB+ IRF4+ precursors to the mature GC B cell state? Theoretically, exposure to IL-21 in the absence of CD40 signaling could promote higher levels of BCL6 without promotion of RelB. IL-21 is required for even intermediate Bcl6 levels in our in vitro culture system, and in vivo B cell expression of IL-21R is required for maximal BCL6 levels in GC B cells (*Linterman et al., 2010*; *Zotos et al., 2010*). However, Tfh cells are known to express CD40L transcripts at levels comparable to Th2 cells (*Nurieva et al., 2008*). Moreover, exposure to IL-21 in the absence of anti-CD40 within our in vitro culture system, even if there was prior CD40 ligation, leads to apoptosis without GC B cell formation, consistent with previous reports of IL-21 induced apoptosis (*Spolski and Leonard, 2008*). Together with the observation that prolonged CD40 agonism inhibits the formation of GC cells, our results suggest that precursor progression critically depends on a temporary abstinence from additional CD40 signaling that would otherwise subvert the second phase of GC B cell maturation. This cannot be explained by an inability of precursor cells to express CD40 or to respond to CD40 ligation, because they remain intrinsically responsive to anti-CD40 both in vitro and in vivo. For these reasons, we favor a model in which the maturation of GC precursor cells instead reflects a transient diminution of cognate T cell engagement.

Our results suggest that a reduction of nuclear RelB and the furthered differentiation of GC precursors is associated with cell division. At a population level both in vitro and in vivo, the emergence of RelB+ IRF4+ BCL6$^{int}$ B cells required multiple cell divisions, and interestingly appeared to be coordinately linked to the number of cell divisions. The Bcl6$^{hi}$ counterparts became evident after an additional cell division. An unequal distribution of BCL6 resulting from asymmetric cell division has previously been reported in a subset of germinal center B cells (*Barnett et al., 2012*). Unfortunately, we are unable to determine whether this occurs during the cell division of GC precursors induced with this model antigen, in part due to the rapid kinetics of cytokinesis, the requirement for real-time imaging of cell division and inadequacy of transcriptional reporters. These putative processes would be ideally defined by in vivo imaging of B cells expressing fusion proteins of these transcription factors.

Together our results are consistent with the idea that attenuated T cell engagement, and presumably fewer cycles of CD40 signaling, encourages the transition of precursors to the BCL6$^{hi}$ state during the early stages of adaptive immune responses. Depleting CD4 T cells in vivo during the critical transitional time period shifted the precursor/GC ratio in favor of GC B cell maturation. Similarly, varying or supplementing antigen dose influenced the timing and extent of GC precursor maturation. This is consistent with prior studies demonstrating that higher affinity B cells with a greater capacity to acquire and present antigen to T cells preferentially formed plasmablasts during early adaptive immune responses (*Schwickert et al., 2011*). In this regard, it is interesting to note that particulate antigen has also been reported to be inequitably distributed between daughters during cell division, such that a subset of B cells receive less antigen and have a reduced capacity to stimulate cognate T cells (*Thaunat et al., 2012*).

A reduced Ag presentation capacity subsequent to multiple cell divisions would be consistent with previous reports of progressively shorter T-B interaction lengths (*Kerfoot et al., 2011*). Similarly, persistent pathogen-derived antigen during chronic *Salmonella* infection results in large numbers of extrafollicular ASCs and impaired GC responses (*Di Niro et al., 2015*). However our results contrast with a previous finding that the introduction of additional Ag potentiates the magnitude of T$_{fh}$ and GC cells (*Baumjohann et al., 2013*). The discrepancy could be explained by their relatively delayed introduction of additional Ag when peri-follicular precursors are no longer present and GC B cells have begun their intra-follicular expansion. As GC precursors differentiate to the Bcl6$^{hi}$ state and move into the follicle interior, a new source of antigen becomes available within the follicular dendritic cell (FDC) network presented in the form of surface retained immune complexes and complement-tagged antigen, a feature unique to this stromal cell type in lymphoid tissue (*Liu et al., 1997*; *Szakal et al., 1989*). Once BCL6$^{hi}$ GC B cells are within the intra-follicular environment, cell division is limited without additional rounds of Tfh engagement. However at this stage they have a rewired transcriptional program unlikely to completely resemble their peri-follicular precursor counterparts, and hence their interactions with and response to Tfh likely differs.

Germinal centers are from an organismal point of view a large investment in space, time and metabolic energy, as well as being mutually exclusive with other urgently needed B cell effectors. It would be strategic to postpone an investment into GCs until the pathogen burden is sensed as diminishing. The results presented here are consistent with the idea that sub-threshold antigen presentation by the precursors to GC B cells attenuates their engagement with cognate T cells during the end of their residence at follicular boundaries. In this regard it is interesting to note that mature GCs harbor a distinct micro-anatomic compartment, the dark zone, that has a paucity of Tfh cells (*Hauser et al., 2010*). It is tempting to speculate that this unique feature of the dark zone, it's dearth of T cell-derived CD40L, contributes to the phased self-renewal of GC B cells by providing an environment that allows them to escape continued T cell engagement.

## Materials and methods

### Mice

B1-8 mice, homozygous for the targeted insertion of the Vh 186.2 Ig heavy chain ($Igh^{VNP/VNP}$) allele derived from the B1-8 hybridoma specific for the hapten NP (*Maruyama et al., 2000*) and also with a homozygous deletion of the Jk locus (*Chen et al., 1993*) were bred locally and used as a source of hapten specific B cells. Ovalbumin-specific TCR-transgenic (OTII) mice (4194; Tg(*TcraTcrb*)425Cbn/J; Jackson) were also bred locally and used as a source of carrier-specific T cells. Hapten or carrier specific mice were further crossed with strains of mice expressing fluorescent proteins within all nucleated cells that produced either dsRed (6051; Tg(CAG-DsRed*MST)1Nagy/J) under control of the $\beta$–Actin promoter or eGFP via the ubiqutin promoter (4353; Tg(UBC-GFP)30Scha/J) obtained from Jackson Laboratory. YFP-BCL6 knock-in reporter mice (*Baumjohann et al., 2011*; *Kitano et al., 2011*) were kindly provided by Dr. K. Mark Ansel, with permission from Dr. Takaharu Okada. This strain harbors a *yfp* construct inserted into exon 1 of the *Bcl6* gene resulting in the expression of hypofunctional chimeric YFP-BCL6 protein. Heterozygous $Bcl6^{yfp/+}$ mice were crossed and genotyped for cell sorting experiments. All mice were housed at the Central Animal Care Facility (Yale University, New Haven) and treated in compliance with the guidelines established by the Yale University Institutional Animal Care and Use Committee (IACUC). No empirical test was used for choosing sample size before experiments. No randomization of samples or animals was used, nor were investigators blinded throughout the study.

### Antibodies

Antibodies with the following specificities used for flow cytometry and histology were purchased from BD Bioscience, eBioscience or Biolegend: CD45.2-biotin, Lamda Light chain-biotin, GFP- Alexa Fluor 488 (AL488), rabbit IgG-PE, CD38-PE-Cy7, CD95-PE-Cy7, BCL6-Al647, Streptavidin-Al700 or -Brilliant Violet 421 (BV421), CD45RA (B220)-allophycocyanin (APC)-Cy7, IRF4-eFluor 450. RelB (C19) was from Santa Cruz Biotech, rabbit IgG Fab$_2$-Al555 was from Cell Signaling Technologies. Purified αCD40 (FGK4.5) (RRID: AB_2490239) and αCD40L (MR1) (RRID: AB_1612465) were purchased from BioXCell. AFRC-Mac-1 cell (glycoprotein of dog chlamydomonas) (RRID: CVCL_K178) (Sigma-Aldrich) produced Rat IgG2a istotype control was purified from culture supernatants by affinity chromatography, using a staphylococcal protein G column (ThermoFisher Scientific) and filter sterilized. Hamster IgG was from Jackson ImmunoResearch Laboratories.

### Adoptive transfers, Immunizations and treatments

Naïve NP-specific B cells or OVA-specific OT-II T cells were isolated from the spleens of B1-8 or OT-II mice by negative selection using EasySep Negative Selection Mouse B or T cell Enrichment Kit (StemCell Technologies). B cells were labeled with 2.5 μM of final concentration CFSE (Invitrogen) following the vendor's protocol. $3 \times 10^6$ Ag-specific B cells together with or without $1.5 \times 10^5$ Ag-specific T cells were transferred into recipient C57BL/6 or CD45.1 congenic mice. 1 day post cell transfer, mice were immunized by footpad (f.p.) injection of 10–15 μg NP-CGG or NP-OVA emulsified in complete Freund's adjuvant (CFA, Sigma) at 1–1.5 μg/μl. For treatment experiments, 20 μg NP-CGG (control PBS), αCD40 (control Rat IgG2a), αCD4 (control Rat IgG2a) and αCD40L (control Hamster Ig) were injected f.p. locally at day 3 after immunization.

## Flow cytometry

Single cell suspensions were pre-incubated with Fc blocking antibody (2.4G2) and stained with LIVE/DEAD Fixable Aqua Dye (Molecular Probes) to discriminate dead cells. After cell surface staining, cells were fixed and permeablized with Cytofix / Cytoperm solution (BD Bioscience) for further intracellular nuclear staining of BCL6, IRF4 and RelB at 4°C overnight. RelB staining was revealed next day by the incubation of Alexa 55–Goat anti-rabbit IgG. Flow cytometry was performed on a FACS LSR II (Becton Dickinson) and analyzed with FlowJo software (TreeStar, Portland, OR).

## Immunofluorescence histology and image analysis

Popliteal lymph nodes were fixed in vitro with 1% paraformaldehyde-lysine-periodate solution, and frozen in OCT (TissueTek) after passage through sucrose gradient solutions. 7 µm–thick cryostat sections were stained with antibody reagents recognizing RelB, BCL6 or IRF4. RelB staining was further revealed by the incubation of Alexa Fluor 555–conjugated anti-rabbit IgG Fab$_2$, or by Streptavidin-BV421 following the incubation of biotinylated anti-rabbit IgG Fab$_2$. After extensive washing, the slides were mounted in Prolong Gold anti-fade reagent (Molecular Probes). Images were taken with an automated wide-field microscope (Nikon Eclipse Ti) and a CCD camera (Qimaging Retiga 2000R) with NIS Elements software. Emitted light was collected through 450/60, 525/50, 605/70, and 700/75 nm bandpass filters. Final processing to overlay single-channel images was performed with Adobe Photoshop. Imaris software (Bitplane/Perkin Elmer) was used to measure the distance between interested cells and the follicle border, which was defined manually in Imaris based on RelB expression, and the mean fluorescence intensity (MFI) of BCL6, RelB and IRF4 in interested cells.

## In vitro cell culture system to generate Bcl6$^{hi}$ RelB$^{lo}$ IRF4$^{lo}$ GC Cells

Freshly isolated primary splenic B cells were co-cultured with a mouse follicular dendritic cell (FDC) derived cell line, FL-YB. FL-YB expresses comparable levels of B cell activating factor (BAFF) to that of FDCs. It was kindly provided by Dr. Masaki Magari and cultured as described in a previous study (*Magari et al., 2011*). FL-YB cells were tested to be mycoplasma free and validated to be overexpressing BAFF that supports murine primary B cell survival. Twenty-four hours before the initiation of cultures, $4 \times 10^3$ FL-YB cells were added into single well of 48-well tissue culture plates to adhere. Purified splenic B cells were first pulse activated with 10 µg/ml goat anti-moue IgM F(ab')$_2$ (Jackson ImmunoResearch Laboratories) as previously described (*Damdinsuren et al., 2010*). αIgM-pulse stimulated B cells ($5 \times 10^5$) in 400 µl culture medium supplemented with 5 µg αCD40 (RRID: AB_2490239)+20 ng/ml IL-21/20 ng/ml IL-4 (PeproTech) were added to FL-YB cell-seeded wells, followed by re-culturing with medium containing αCD40 + IL-21/4 2 days later. At 72 hr, cells were washed and centrifuged in order to completely remove stimuli. Finally the cells were re-cultured with medium only for extra half-day in FL-YB-seeded wells.

## Detection of Bcl6 RNA by flow cytometry

Primary splenic B cells were cultured as described above. At the end of culture, cells were harvested and mRNA expression of *bcl6* was analyzed on a single cell level by flow cytometry in combination with surface markers B220, CD38 and intracellular BCL6, RelB and IRF4 protein staining, using a PrimeFlow RNA Assay kit (Affymetrix eBioscience, catalog #VB1-15400-204) according to manufacturer's protocols with minor revision. Intracellular protein staining was carried out at 4°C overnight in the presence of RNAase inhibitors. Type 1 probe set (Alexa Fluor 647) was chosen for mouse *bcl6* or positive control *actb*. Flow cytometry was performed on a FACS LSR II (Becton Dickinson) and analyzed with FlowJo software (TreeStar, Portland, OR).

## Cell sorting

Primary splenic *Bcl6$^{yfp/+}$* B cells were cultured as described above. About 62 hr after culture, cells were harvested and surface stained for CD38, CD86 and B220 with discrimination of dead cells. YFP-Bcl6$^{hi}$ CD38$^{hi}$ CD86$^{hi}$ and YFP-Bcl6$^{lo}$ CD38$^{hi}$ CD86$^{hi}$ cells were sorted with a BD FACSAria II. Sorted cells were re-cultured with medium only for an additional 12 hr in FL-YB-seeded wells.

## Statistical analysis

Prism software (Graphpad) was used to graph the data and for calculation of statistical significance. Comparisons between two independent groups were assessed by unpaired t-tests or multiple t-tests (different fraction of group); between two dependent groups were assessed by paired t-tests. Multiple comparisons were assessed by one-way ANOVA. A Repeated Measures ANOVA was used for comparison of different fraction within group. A P value less than 0.05 was considered statistically significant.

## Acknowledgements

We thank Dr. Aaron J Marshall for critical review of the manuscript and Dr. K Mark Ansel for kindly providing the YFP-BCL6 knock-in reporter mice with permission from Dr. Takahura Okada. This work was supported by NIH R01AI080850 and R21AI101704 grants to AMH, and TTZ was supported by a Canadian Institute of Health Research Postdoctoral Fellowship.

## Additional information

### Funding

| Funder | Grant reference number | Author |
| --- | --- | --- |
| National Institutes of Health | R01AI080850 | Ann M Haberman |
| National Institutes of Health | R21AI101704 | Ann M Haberman |
| Canadian Institutes of Health Research | Postdoctoral fellowship | Ting-ting Zhang |

The funders had no role in study design, data collection and interpretation, or the decision to submit the work for publication.

### Author contributions

T-tZ, Conceptualization, Data curation, Formal analysis, Methodology, Writing—original draft, Writing—review and editing; DGG, Resources, Data curation, Methodology; CMC, SD, YC, Data curation, Methodology; SMK, Resources, Methodology, Writing—review and editing; MM, Resources, Writing—review and editing; AMH, Conceptualization, Resources, Funding acquisition, Validation, Methodology, Writing—review and editing

### Author ORCIDs

Ting-ting Zhang, http://orcid.org/0000-0002-7868-9624
Ann M Haberman, http://orcid.org/0000-0001-5168-8229

### Ethics

Animal experimentation: This study was performed in strict accordance with the recommendations in the Guide for the Care and Use of Laboratory Animals of the National Institute of Health. All of the animals were handled according to approved institutional animal care and use committee (IACUC) protocols (#2016-11326) of Yale University. Yale University's Animal Welfare Assurance number is #A3230-01.

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
