## [Decision Letter]

[Editors’ note: this article was originally rejected after discussions between the reviewers, but the authors were invited to resubmit after an appeal against the decision.]

Thank you for submitting your work entitled "GC B cell development has distinctly regulated stages completed by disengagement from T cell help" for consideration by *eLife*. Your article has been favorably evaluated by Arup Chakraborty (Senior Editor) and three reviewers, one of whom is a member of our Board of Reviewing Editors.

Our decision has been reached after consultation between the reviewers. Based on these discussions (which are critically important) and the individual reviews below, we regret to inform you that your work will not be considered further for publication in *eLife*.

In this manuscript, authors asked the longstanding important question of how pre-GC B cells make fate decisions such as differentiation into GC B cells versus plasmablasts. One of the most important conclusions is that Bcl6^int^ IRF4^+^ cells are the precursor of Bcl6^hi^ IRF4^lo^ GC B cells. And, their observations by themselves are of potential interest.

However, as is clear from the reviewers' comments (appended below), the most critical concern of all reviewers is that this conclusion is not warranted without direct transfer experiments. Since you identify this precursor fraction by FACS using permeablized cells, this method cannot be simply applied for the transfer experiments. But, in principle, the Bcl6^int^ IRF4^+^ cells could be purified by employing other methods such as using Bcl6 reporter mice or surrogate surface markers, thereby allowing you to address this issue. However, such experiments would require a major effort.

*Reviewer #1:*

This manuscript has addressed the mechanisms underlying the fate decisions of pre-GC B cells, particularly differentiation into GC cells. In this regard, Brink's group initially demonstrated (JEM; 2006) that high affinity cells differentiate to plasmablasts, whereas lower affinity cells go to GC cells, thereby concluding that affinity is an instructive signal. In his group's consecutive paper (JI; 2009), they showed that affinity was promoting plasmablast expansion rather than directing the differentiation. Therefore, how to make decisions between GC and plasmablasts at this pre-GC phase is an important issue, but being still controversial.

Authors used Bcl6, IRF4, RelB as a marker for tracing the status of B cells, and employed in vivo as well as in vitro experiments. Then, they conclude that CD40 is required to generate the precursor of GC B cells, but their transition to mature GC cells is promoted by diminution of T cell help. I think that overall experimental designs are very good, and the conclusion is well guaranteed at this stage (although I have a couple of concerns as described below). Particularly, the novel points of this paper are to distinguish precursor GC and mature GC by using several markers, and to determine the effects of perturbation of T cell help and CD40 signal on precursor GC and mature GC.

My concerns:

1) Since authors utilized in vivo expression levels of RelB as a marker of CD40 signaling, they should show that this assay is trustable enough in vivo. Although previous papers (JBC;2008, JEM;2008, ARI; 2009) showed correlation between CD40 signal and RelB protein expression, this is just in vitro correlation. Two questions; 1) Do other signals such as BCR, cytokine and BAFF affect the expression of RelB in vivo? 2) How about quantity control? If, in vivo expression of RelB is weak, can we consider low level of in vivo CD40 signals?

2) In Figure 2, not only blue fraction (Bcl6^hi^ IRF4^lo^) but black fraction (Bcl6^lo^IRF4^hi^) appears to be differentiated from the GC precursors. Is the black fraction pre-plasmablasts? Characterization of this black fraction is not a main topic in this manuscript. But, considering that differentiation into GC or plasmablast is a key issue and that in Fig, 6, authors have assayed differentiation into plasma cells, too, some characterization of the black fraction could be well connected to the Figure 6 data, thereby providing a more insight into this manuscript.

3) Since this manuscript addresses the pre-GC phase, extrapolating their results into events occurring during GC cycles is an overstatement. In this regard, I feel that the last paragraph in the Discussion (particularly from the sentence starting “It is unclear whether a transient …” to the end) is not appropriate. For instance, Bannard paper shows similar proliferation rate between wt and CXCR4 ko, although in the long term, ko was outcompeted.

*Reviewer #2:*

Zhang et al. addressed the long-standing question in B cell biology about the roles for CD40 signaling, which is required for GC B cell development but paradoxically inhibits Bcl6 expression. From the results of the in vivo and in vitro experiments, the two-step model of GC B cell differentiation is proposed as an answer to the above question. In this model, Bcl6^int^ IRF4^int^ GC precursors are formed in a CD40 signaling-dependent manner, and then the precursors become Bcl6^hi^ IRF4^lo^ GC B cells in a manner dependent on the attenuation of CD40 signaling. Overall, the experiments appear well designed and performed (especially the flow analysis of Bcl6 and IRF4), the results seem convincing (especially the effects of the in vivo treatments for 0.5 day), and the interpretations read mostly reasonable to justify the proposed model. By experimentally addressing the four points described below, the authors could strengthen the justification of the model and give more mechanistic insights into the steps of Bcl6 upregulation.

1) Although it seems most likely that Bcl6^int^ IRF4^int^ B cells are really the precursor of Bcl6^hi^ IRF4^lo^ GC B cells, there is no direct evidence for this. The authors could address this by sorting Bcl6^int^ IRF4^int^ B cells and Bcl6^lo^ IRF4^lo^ B cells from the culture system or from immunized mice and putting them in the culture. If Bcl6^int^ IRF4^int^ B cells become Bcl6^hi^ IRF4^lo^ B cells more quickly and/or more efficiently than Bcl6^lo^ IRF4^lo^ do, that would be the best compelling evidence. The culture system used in this work is really nice. However, without the sorting experiments, the in vitro results do not provide insights that the in vivo experiments could not.

2) The specificity of Bcl6 and IRF4 staining in the histological analysis need to be shown. It is particularly important to show the specificity of intermediate Bcl6 staining. Ideally, the use of knockout B cells would be preferable, but the anti-CD40 ligand treatment shown in Figure 1—figure supplement 1 might be used because it should strongly diminish not only RelB signals but also intermediate Bcl6 and IRF4 upregulation at 3 days after immunization.

3) Nuclear vs. cytosolic staining of RelB is unclear in many cells. For example, there are two Bcl6^+^"RelB^-^" cells with clear RelB signals in Figure 7 (the right circle in the top row and the far left circle in the bottom row). Are the authors claiming that they are cytosolic signals? If so, in Figure 1 there are several "RelB^+^" cells (polygon and square) in which RelB signals seem cytosolic. The author should show their criteria for nuclear staining using magnified images of individual cells.

4) It is known that Bcl6^hi^ GC B cells express upregulated amounts of both mRNA and protein of Bcl6. What about Bcl6int precursors? Do they, too, have elevated levels of Bcl6 mRNA, or only the protein expression is increased compared to activated Bcl6^lo^ B cells or naïve B cells? Allman et al. (Blood, 1996) showed that CD40 signaling quickly reduced Bcl6 mRNA expression in B cells. It would be consistent if the Bcl6^int^ precursors upregulated Bcl6 protein but not mRNA.

*Reviewer #3:*

This reviewer's major concern with this manuscript is their central premise that Bcl6^int^ Relb^+^ cells are the "precursors" of GC B cells that are Bcl6^hi^ Relb^-^.The authors base this claim on the following pieces of evidence:

1) The appearance of Bcl6^int^ Relb^+^ cells temporally precedes that of the Bcl6^hi^ Relb^-^ cells

2) Bcl6^int^ Relb^+^ cells are to a greater extent represented among cells that have divided fewer times than Bcl6^hi^ Relb^-^ cells, which have divided more times.

3) These two cell types bear somewhat overlapping surface phenotypes by flow cytometry.

However, as the authors show in Figure 2, the Bcl6^int^ Relb^+^ cells comprise only 10% of the total transferred responding cells. Thus, on day 4.5, when the authors show that the Bcl6^hi^ Relb^-^ cells begin appearing, how do they know that these cells derive from the 10% of cells that were Bcl6^int^ Relb^+^ cells, or rather that they derive from the ~90% remainder of transferred cells that are Bcl6^-^ Relb^-^? The above 3 lines of evidence do not bear on this question, making the central premise of this manuscript fundamentally unsupported by the evidence presented.

The only acceptable evidence in this regard that would substantiate such a claim would be direct transfer of purified Bcl6^int^ Relb^+^ cells into immunization-matched recipient mice and then following the fate of these cells.

All other interpretations of the subsequent experiments (varying immunogen dose, CD40 antibody treatment, etc.) are based on this assumption, since the authors use Bcl6^int^ Relb^+^ cells as a read out of their various manipulations. Nevertheless, even if one accepts the authors' premise of these cells as the precursors of GC cells, there are several concerns with these experiments as well:

1) Varying the immunogen dose and reading out the Bcl6^int^ Relb^+^ cells and Bcl6^hi^ Relb^-^ cells is not mechanistically interpretable (Figure 5). When B cells are provided more antigen, they will in theory signal to a greater extent through their BCR and present more peptide-MHC to T cells. Thus, this experiment on its own is uninterpretable, as any difference in outcome obtained by these means can be attributable to either BCR or T cell help.

2) In Figure 6, the authors attempt to test the effect of "sustained T cell help." The first way in which they do this is by injecting soluble antigen on day 3 of the response. The authors observe fewer GC cells and more Bcl6^int^ Relb^+^ cells after soluble NP-CGG injection. It's unclear what this represents however because B cells do not acutely encounter a large amount of antigen in the soluble form in physiological responses. Moreover, as Shokat et al., Nature 1995 showed, acute injections of soluble antigen rapidly induce the death of GC B cells. These previous observations may account for the present authors' findings. The effect of soluble antigen on increasing Bcl6^int^ Relb^+^ cells is unclear, given that it is not clear whether these are in fact only precursors of GC cells. Finally, CD40 antibody treatment is an imprecise and non-physiological means of mimicking "persistent T cell help" given the constant presence of antibody stimulation, in contrast to the frequent, yet intermittent, T-B contacts at this stage of the response. Finally, the notion that increased T cell help increases plasmablast production has already been shown by Schwickert et al., JEM 2011, which used a more physiological method to increase peptide-MHC on B cells and increasing their resulting T cell help. This should be acknowledged by these authors.

3) The experiments in Figure 7 are puzzling. Blocking CD40 signaling has long been known to impair GC formation (for example, Han et al., Journal of Immunology 1995). The authors suggest that 0.5 days after injecting anti-CD40, they see fewer Bcl6^int^ Relb^+^ cells but unchanged numbers of GC cells. Based on this, the authors conclude that this "promotes this transition" from precursors to GC cells at this early stage. This reviewer does not agree that such a conclusion can be made from this experiment. As previously mentioned, the precursor-product relationship of these cells is far from clear. Furthermore, the authors claim that Bcl6^int^ Relb^+^ cells decline but are not undergoing more apoptosis, and suggest that they therefore must be transitioning instead into GC cells. However, ex vivo caspase stains such as the one used are notoriously unreliable measures of the true rates of cell death in vivo – as Figure 7 illustrates, only 0.27% of cells are undergoing apoptosis, a number that is incredulously low.

Other concerns:

1) The authors mention that "persistent CD40 signaling is not seen in GC B cells" and cite Basso et al. 2004. This is not an accurate statement. Victora et al., Cell 2010 performed gene expression analysis on light zone GC B cells. They indeed found an enriched signature of CD40 signaling among these cells. This should be referenced by the authors.

2) Several inappropriate references to Kerfoot et al. 2011 are made in the Introduction. This was not the original work that delineated the cellular interactions between T and B cells at the T-B border that precede the GC phase of the antibody response. Rather, these findings were made by Garside et al. and Okada et al., Science, 1998 and Okada et al., PLoS Biology, 2005.

In summary, the conclusions made by Zhang et al. are not supported by the evidence.

---

## [Author Response]

[Editors’ note: the author responses to the first round of peer review follow.]

*In this manuscript, authors asked the longstanding important question of how pre-GC B cells make fate decisions such as differentiation into GC B cells versus plasmablasts. One of the most important conclusions is that Bcl6^int^ IRF4^+^ cells are the precursor of Bcl6^hi^ IRF4^lo^ GC B cells. And, their observations by themselves are of potential interest.*

*However, as is clear from the reviewers' comments (appended below), the most critical concern of all reviewers is that this conclusion is not warranted without direct transfer experiments. Since you identify this precursor fraction by FACS using permeablized cells, this method cannot be simply applied for the transfer experiments. But, in principle, the Bcl6^int^ IRF4^+^ cells could be purified by employing other methods such as using Bcl6 reporter mice or surrogate surface markers, thereby allowing you to address this issue. However, such experiments would require a major effort.*

*Reviewer #1:*

*[…] My concerns:*

*1) Since authors utilized in vivo expression levels of RelB as a marker of CD40 signaling, they should show that this assay is trustable enough in vivo. Although previous papers (JBC;2008, JEM;2008, ARI; 2009) showed correlation between CD40 signal and RelB protein expression, this is just in vitro correlation.*

The point raised regarding the specificity of RelB as a marker of CD40 signaling in B cells is important. We have carefully examined it in vivo using CD40 ligand blockade (Figure 1—figure supplement 1). We also tested by histology another NFkB molecule, p52, that is also known to play a role in the canonical pathway downstream of the BCR. As expected, RelB levels are quite reliant on CD40 signaling, in contrast to p52 expression in Ag-specific B cells (Figure 9).

Author response image 1.**DOI:**
http://dx.doi.org/10.7554/eLife.19552.018

*Two questions; 1) Do other signals such as BCR, cytokine and BAFF affect the expression of RelB in vivo? 2) How about quantity control? If, in vivo expression of RelB is weak, can we consider low level of in vivo CD40 signals?*

1) BCR is known to activate the canonical (classical) pathway of NF-κB through multiple mechanisms, but not the non-canonical NF-κB pathway that employs RelB (Annu. Rev. Immunol. 2009 by Vallabhapurapu S et al.). Lymphotoxin (LT) can activate the non-canonical NF-κB pathway, but the receptor for LT is not expressed in B cells. We are aware that BAFF is secreted by macrophages, DCs and stromal cells, and can activate the non-canonical NF-κB pathway in B cells, known to be critical for mature B cell survival. However, the lack of RelB expression (in contrast to p52) in non-responding follicular B cells (Figure 1—figure supplement 1) indicates that RelB is less affected by BAFF in vivo. Our in vitro culture data also suggest that RelB expression levels are less reliant on the BAFF produced by the FDC-derived feeder cell line than the introduced anti-CD40.

2) Throughout the study, we judged the positive signal of RelB based on a threshold set by non-responding B cells, which are lack of RelB expression (Figure 1—figure supplement 1). If the RelB signal is above the threshold but at a low level, we still consider it indicative of a low level of CD40 signaling, as shown by the MFI of RelB in Figure 4, Figure 5 and 6.

*2) In Figure 2, not only blue fraction (Bcl6^hi^ IRF4^lo^) but black fraction (Bcl6^lo^IRF4^hi^) appears to be differentiated from the GC precursors. Is the black fraction pre-plasmablasts? Characterization of this black fraction is not a main topic in this manuscript. But, considering that differentiation into GC or plasmablast is a key issue and that in Fig, 6, authors have assayed differentiation into plasma cells, too, some characterization of the black fraction could be well connected to the Figure 6 data, thereby providing a more insight into this manuscript.*

We had debated whether we should further discuss the Bcl6^lo^ IRF4^hi^ cells that are also prominent. We feared this could confound the main topic of this manuscript. Bcl6^lo^ IRF4^hi^ cells have been characterized as plasmablasts by Kyoko Ochiai et al. (2013 Immunity). Based on the earlier appearance of this subset and cell division data, the Bcl6^lo^ IRF4^hi^ population appeared less reliant on GC precursors. The BCL6^lo^ IRF4^hi^ cells that were evident during early generations continued to expand as aggressively as GC B cells towards later cell divisions. We have incorporated the data into Figure 4—figure supplement 1 and added a sentence in the Results section (subsection “The coordinated transition of GC precursors to BCL6^hi^ GC B cells is associated with cell division and a loss of RelB”, second paragraph).

*3) Since this manuscript addresses the pre-GC phase, extrapolating their results into events occurring during GC cycles is an overstatement. In this regard, I feel that the last paragraph in the Discussion (particularly from the sentence starting “It is unclear whether a transient …” to the end) is not appropriate. For instance, Bannard paper shows similar proliferation rate between wt and CXCR4 ko, although in the long term, ko was outcompeted.*

This discussion was of course speculative and was designed to provide the reader with a broader perspective on how these molecular processes might also play a role in maintenance of established GCs. We have removed the discussion of the Bannard et al. paper and have instead merely drawn attention to the dark zone compartment’s paucity of T cells (Discussion, last paragraph).

*Reviewer #2:*

*Zhang et al. addressed the long-standing question in B cell biology about the roles for CD40 signaling, which is required for GC B cell development but paradoxically inhibits Bcl6 expression. From the results of the in vivo and in vitro experiments, the two-step model of GC B cell differentiation is proposed as an answer to the above question. In this model, Bcl6^int^ IRF4^int^ GC precursors are formed in a CD40 signaling-dependent manner, and then the precursors become Bcl6^hi^ IRF4^lo^ GC B cells in a manner dependent on the attenuation of CD40 signaling. Overall, the experiments appear well designed and performed (especially the flow analysis of Bcl6 and IRF4), the results seem convincing (especially the effects of the in vivo treatments for 0.5 day), and the interpretations read mostly reasonable to justify the proposed model. By experimentally addressing the four points described below, the authors could strengthen the justification of the model and give more mechanistic insights into the steps of Bcl6 upregulation.*

*1) Although it seems most likely that Bcl6^int^ IRF4^int^ B cells are really the precursor of Bcl6^hi^ IRF4^lo^ GC B cells, there is no direct evidence for this. The authors could address this by sorting Bcl6^int^ IRF4^int^ B cells and Bcl6^lo^ IRF4^lo^ B cells from the culture system or from immunized mice and putting them in the culture. If Bcl6^int^ IRF4^int^ B cells become Bcl6^hi^ IRF4^lo^ B cells more quickly and/or more efficiently than Bcl6^lo^ IRF4^lo^ do, that would be the best compelling evidence. The culture system used in this work is really nice. However, without the sorting experiments, the in vitro results do not provide insights that the in vivo experiments could not.*

The issue raised by the reviewer is very insightful and constructive. We have performed experiments to further test the GC precursor-product relationship using purified precursor cells. For this we have employed a strain of mice expressing a Bcl6-YFP fusion protein under the direction of the Bcl6 promoter (Kitano M et al. Immunity 2011; Baumjohann D et al. JI 2011). By using a stringent gate for YFP combined with CD38 expression, a gene repressed by Bcl6 in mature GC B cells, and CD86 (up-regulated with NFκB activation). we have sorted in vitro cultured cells for YFP+ CD38^hi^ CD86^hi^ cells that are highly enriched for Bcl6int IRF4^+^ RelB^+^ precursor cells. After they were returned to culture under permissive conditions (CD40/cytokine washout), only the YFP^hi^ population, but not the YFP^lo^ population, gave rise to BCL6 hi IRF4 lo RelB lo cells (Figure 8—figure supplement 2). These results are consistent with the proposed precursor-product relationship. Although the rate of differentiation was lower than we had hoped, it is likely that the long duration of cell staining and sorting process compromised their further differentiation during the subsequent 12 hour re-culture. We have added relevant information in the Results section (subsection “Deprivation of T cell derived stimuli promotes the in-vitro generation of Bcl6^hi^ IRF4^-^ RelB^-^ B cells”, fourth paragraph) with Figure 8—figure supplement 2, and in Materials and methods section (subsection “Detection of Bcl6 RNA by flow cytometry”).

*2) The specificity of Bcl6 and IRF4 staining in the histological analysis need to be shown. It is particularly important to show the specificity of intermediate Bcl6 staining. Ideally, the use of knockout B cells would be preferable, but the anti-CD40 ligand treatment shown in Figure 1—figure supplement 1 might be used because it should strongly diminish not only RelB signals but also intermediate Bcl6 and IRF4 upregulation at 3 days after immunization.*

As suggested by the reviewer, we have included additional data in Figure 1—figure supplement 3 showing that after CD40L blockade in vivo, Bcl6 expression by B cells is not apparent (subsection “Intermediate levels of BCL6 are found in a subset of RelB^+^ IRF4^+^ Ag-specific B cells prior to the emergence of follicular BCL6^hi^ GC B cells that lack RelB and IRF4”, last paragraph). The specificity of IRF4 staining has been confirmed elsewhere (Kyoko Ochiai et al. 2013 Immunity), and it is upregulated by either BCR signaling and /or CD40 activation. We found that blockage of CD40L does not completely diminish IRF4 expression (Figure 10).

Author response image 2.**DOI:**
http://dx.doi.org/10.7554/eLife.19552.019

*3) Nuclear vs. cytosolic staining of RelB is unclear in many cells. For example, there are two Bcl6^+^"RelB^-^" cells with clear RelB signals in Figure 7 (the right circle in the top row and the far left circle in the bottom row). Are the authors claiming that they are cytosolic signals? If so, in Figure 1 there are several "RelB^+^" cells (polygon and square) in which RelB signals seem cytosolic. The author should show their criteria for nuclear staining using magnified images of individual cells.*

We appreciate the issue raised the reviewer. We did observe that most activated B cells express RelB in the nuclear form, but some cells did express RelB in the cytosolic form. The later may be indicative of recent ligation of CD40 or cells that are active in the cell cycle after CD40 ligation. For the identification of RelB^+^ cells, we had been less stringent in Figure 1 by including all the transferred B cells with clear RelB signal without discrimination of nuclear vs cytosolic. For Figure 7, we were more stringent with the selection criteria by comparing RelB expression vs nuclear staining of Bcl6. The point raised is well taken – we should be more consistent with our selection criteria. Therefore, we have applied the more stringent selection criteria for nuclear RelB as RelB^+^ to the data in Figure 1, and re-analyzed image data for the distance (Figure 1) and the MFI (Figure 1). We found that this did not change the results or the significance. In regard to the unclear staining pattern of RelB in some cells, we now show magnified images of individual cells with RelB and nuclear staining of Bcl6 in a Figure 1—figure supplement 2.

*4) It is known that Bcl6^hi^ GC B cells express upregulated amounts of both mRNA and protein of Bcl6. What about Bcl6int precursors? Do they, too, have elevated levels of Bcl6 mRNA, or only the protein expression is increased compared to activated Bcl6^lo^ B cells or naïve B cells? Allman et al. (Blood, 1996) showed that CD40 signaling quickly reduced Bcl6 mRNA expression in B cells. It would be consistent if the Bcl6^int^ precursors upregulated Bcl6 protein but not mRNA.*

This is an insightful and exciting question and a great suggestion! Because the process of staining and sorting cells can lead to the death of Bcl6^hi^ GC B cells due to their predisposition to apoptosis, we pursued this using an in-situ hybridization approach compatible with flow cytometry (PrimeFlow, Affymetrix) (details see Materials and methods subsection “Detection of Bcl6 RNA by flow cytometry”). Although there are some limitations in the dyes that can be used, we are quite pleased with the results. From this analysis we conclude that transcriptional regulation is primarily responsible for the increase in Bcl6 protein levels in Bcl6^int^ precursor cells, but that both transcriptional and post-transcriptional regulation is involved in the subsequent transition to the Bcl6^hi^ GC B cell state (Figure 8 and Figure 8—figure supplement 1) (added in Results subsection “Deprivation of T cell derived stimuli promotes the in-vitro generation of Bcl6^hi^ IRF4^-^ RelB^-^ B cells”, third paragraph; and also in Discussion section, second paragraph). We thank the reviewer for this excellent suggestion that has resulted in additional information to this differentiative process. To be clear however, we are unable to pursue further the mechanistic basis for the post-transcriptional regulation, which is clearly outside the scope of this study.

*Reviewer #3:*

*This reviewer's major concern with this manuscript is their central premise that Bcl6^int^ Relb^+^ cells are the "precursors" of GC B cells that are Bcl6^hi^ Relb^-^. The authors base this claim on the following pieces of evidence:*

*1) The appearance of Bcl6^int^ Relb^+^ cells temporally precedes that of the Bcl6^hi^ Relb^-^ cells*

*2) Bcl6^int^ Relb^+^ cells are to a greater extent represented among cells that have divided fewer times than Bcl6^hi^ Relb^-^ cells, which have divided more times.*

*3) These two cell types bear somewhat overlapping surface phenotypes by flow cytometry.*

*However, as the authors show in Figure 2, the Bcl6^int^ Relb^+^ cells comprise only 10% of the total transferred responding cells. Thus, on day 4.5, when the authors show that the Bcl6^hi^ Relb^-^ cells begin appearing, how do they know that these cells derive from the 10% of cells that were Bcl6^int^ Relb^+^ cells, or rather that they derive from the ~90% remainder of transferred cells that are Bcl6^-^ Relb^-^? The above 3 lines of evidence do not bear on this question, making the central premise of this manuscript fundamentally unsupported by the evidence presented.*

*The only acceptable evidence in this regard that would substantiate such a claim would be direct transfer of purified Bcl6^int^ Relb^+^ cells into immunization-matched recipient mice and then following the fate of these cells.*

As mentioned in the manuscript, we argue against the idea that Bcl6-lo RelB-lo cells directly give rise to Bcl6-hi RelB-lo cells. Presumably the reviewer is envisioning a CD40L independent mechanism for this hypothetical transition. Our results indicate that CD40L blockade 1 day after immunization both diminishes the number of Bcl6-int Relb^+^ cells and prevents the formation of Bcl6 hi RelB lo cells. This suggests that Bcl6-lo RelB-lo cells cannot directly become Bcl6-hi Rel-lo cells without passage through a CD40L-dependent phase.

We hoped to address reviewer concerns about the precursor-product relationship using purified B cell subsets taken from Bcl6-YFP reporter mice (described above) and a direct transfer to recipients, as suggested. A couple of technical limitations and a theoretical caveat prevented us from taking that approach, namely the predisposition of Bcl6 expressing cells to apoptosis during sorting and the tendency of injected cells to first collect in the lungs and to vary in their arrival to lymphoid tissue. The end assay would involve a collection of those same cells roughly 12 hours after transfer, but most of the donor cells would not have resided in relevant tissue for much of that time period.

We instead employed the in vitro culture system as a source of activated B cell subsets and sorted them based on their YFP and CD38 levels followed by additional short-term culture for further differentiation. The results show that the YFP^hi^ population, but not the YFP^lo^ population, gave rise to BCL6 hi IRF4 lo RelB lo cells (Figure 8—figure supplement 2) (added in Results subsection “Deprivation of T cell derived stimuli promotes the in-vitro generation of Bcl6^hi^ IRF4^-^ RelB^-^ B cells”, fourth paragraph). These results are consistent with the proposed precursor-product relationship. Although the rate of differentiation was lower than we had hoped, it is likely that the long duration of cell staining and sorting process compromised their further differentiation during the subsequent 12 hour re-culture.

*All other interpretations of the subsequent experiments (varying immunogen dose, CD40 antibody treatment, etc.) are based on this assumption, since the authors use Bcl6^int^ Relb^+^ cells as a read out of their various manipulations. Nevertheless, even if one accepts the authors' premise of these cells as the precursors of GC cells, there are several concerns with these experiments as well:*

*1) Varying the immunogen dose and reading out the Bcl6^int^ Relb^+^ cells and Bcl6^hi^ Relb^-^ cells is not mechanistically interpretable (Figure 5). When B cells are provided more antigen, they will in theory signal to a greater extent through their BCR and present more peptide-MHC to T cells. Thus, this experiment on its own is uninterpretable, as any difference in outcome obtained by these means can be attributable to either BCR or T cell help.*

We agree that varying immunogen dose could also affect the extent of BCR crosslinking. BCR signaling and subsequent Ag presentation for T cell help has been difficult to separate for their individual effects. However, we did not intend on having this experiment stand alone as sole proof of our working hypothesis. Instead, it was designed to compliment the other experimental approaches. We modified the text to further make clear that these results are only consistent with the working hypothesis, but does not prove it (subsection “Sustained T cell help at the initiation stage diverts the fate of GC precursors towards plasmablast formation”).

*2) In Figure 6, the authors attempt to test the effect of "sustained T cell help." The first way in which they do this is by injecting soluble antigen on day 3 of the response. The authors observe fewer GC cells and more Bcl6^int^ Relb^+^ cells after soluble NP-CGG injection. It's unclear what this represents however because B cells do not acutely encounter a large amount of antigen in the soluble form in physiological responses. Moreover, as Shokat et al., Nature 1995 showed, acute injections of soluble antigen rapidly induce the death of GC B cells. These previous observations may account for the present authors' findings. The effect of soluble antigen on increasing Bcl6^int^ Relb^+^ cells is unclear, given that it is not clear whether these are in fact only precursors of GC cells. Finally, CD40 antibody treatment is an imprecise and non-physiological means of mimicking "persistent T cell help" given the constant presence of antibody stimulation, in contrast to the frequent, yet intermittent, T-B contacts at this stage of the response. Finally, the notion that increased T cell help increases plasmablast production has already been shown by Schwickert et al., JEM 2011, which used a more physiological method to increase peptide-MHC on B cells and increasing their resulting T cell help. This should be acknowledged by these authors.*

We agree with the reviewer that injecting either model antigens or agonistic aCD40 is not a physiological way to provide persistent T cell help in vivo. Few experimental systems are irreproachably physiologic. We presume that the reviewer’s concern about “soluble” antigen reflects a concern about BCR crosslinking in the absence of adjuvant. The Shokat et al. study mentioned by the reviewer was instead designed to test the impact of BCR crosslinking on mature GC B cells in the *absence* of T cell help, a possible means by which self-reactive B cells could be eliminated. In that study, HEL-specific B cells were transferred into DEL-primed mice and then boosted with presumably soluble DEL to encourage the transferred B cells to enter into GCs at d12. It is important to note that a boost of HEL (with 1000-fold higher binding affinity than DEL) did not induce the transferred cells to proliferate because of the lack of T cell help capable of responding to the amino acid difference in the HEL epitope. They found that injection of an extremely large quantity of HEL (5mg) induced rapid apoptosis of HEL-binding GC B cells at d12. The authors concluded that this may represent a mechanism to eliminate self-reactive B cells that lacked T cell help. Thus, the Shokat study differs significantly from ours in that abundant T cell help is present in our system at the time the mice were boosted with 20 ug of soluble NP-OVA, a quantity comparable to the original immunization. Moreover, continuous injection of soluble Ag has been used previously to sustain T follicular helper cell responses, without evidence of an acutely induced cell death (Baumjohann D et al. 2013 Immunity).

Although the administration of agonistic aCD40 is not ideal, it has been a generally accepted method to mimic one type of T cell help both in vivo and in vitro. The ligated CD40 on the surface of B cells is internalized and in this way would have a limited signaling duration. Admittedly, this cannot perfectly simulate CD40 signaling during T-B encounters, because of course many other signaling pathways are simultaneously invoked during immunological synapse formation of variable durations. It is however an acceptable approach to discreetly mimic this one molecular event in the absence of others.

As the reviewer mentioned, there is an alternative way to deliver antigen to B cells that by-steps BCR crosslinking. The anti-DEC205 approach described by Schwickert et al., JEM 2011 can cause B cells to internalize targeted antigens, but it has its own underappreciated caveats as DEC205 ligation is known to directly and independently induce proliferation. The strain of mice required for DEC205 targeting in B cells is not available to the research community. As suggested, we cite this study along with the other studies reporting the impact of increased T cell help.

*3) The experiments in Figure 7 are puzzling. Blocking CD40 signaling has long been known to impair GC formation (for example, Han et al., Journal of Immunology 1995). The authors suggest that 0.5 days after injecting anti-CD40, they see fewer Bcl6^int^ Relb^+^ cells but unchanged numbers of GC cells. Based on this, the authors conclude that this "promotes this transition" from precursors to GC cells at this early stage. This reviewer does not agree that such a conclusion can be made from this experiment. As previously mentioned, the precursor-product relationship of these cells is far from clear. Furthermore, the authors claim that Bcl6^int^ Relb^+^ cells decline but are not undergoing more apoptosis, and suggest that they therefore must be transitioning instead into GC cells. However, ex vivo caspase stains such as the one used are notoriously unreliable measures of the true rates of cell death in vivo – as Figure 7 illustrates, only 0.27% of cells are undergoing apoptosis, a number that is incredulously low.*

To address one concern highlighted here, we would like to remind the reviewer that chronic and long-term CD40 signaling blockade in our system also impairs GC numbers at later time points (currently shown in a supplemental figure). The goal of the study presented in this manuscript was to examine the factors that influence peri-follicular differentiation that occurs with a very narrow window of time. We approached this using a short-term blockade of CD40 ligation (d3 to d3.5 post-immunization) and an assessment very shortly thereafter. This allowed us to observe stages of GC B cell development distinctly regulated by CD40 signaling that would otherwise not be discernable after protracted blockade.

The low rate of detected apoptosis among the precursors to GC B cells is questioned by the reviewer as being lower than their envisioned death rate for activated B cells during the earliest stages of an immune response. We would like to point out that mature GC B cells within established GCs have a very different transcriptional program that is uniquely predisposed to apoptosis ex vivo. The same cannot be presumed of precursor GC B cells a priori. Although there are no published reports of apoptotic rates for such an early stage of differentiation in vivo, there are numerous reports that examined fully established GCs. Active caspase 3 is apparent in around 5-10% of mature (d10-12) GC B cells (references: Ahmad Zaheen et al., Blood 2009; Sally E. Trabucco et al., JI 2016). Although we have not assessed the apoptotic rate in older GCs in this experimental system, at d4 and d6 we found a somewhat lower percentage of aCasp3+ GC B cells (2-4%, Figure 11).

Author response image 3.**DOI:**
http://dx.doi.org/10.7554/eLife.19552.020

In the literature, the percentage of apoptotic cells in mature GC B cells detected by active caspase 3 is consistent with the values observed using alternative approaches including TUNEL staining (Ref: Sally E. Trabucco et al., JI 2016) and CaspGLOW staining (Ref: Kin L. Good-Jacobson, JI 2012). TUNNEL staining of tissue sections has the advantage of detecting apoptotic nuclei engulfed by phagocytes but not yet degraded. To further validate our active caspase 3 staining, we have included additional negative and positive controls including a short-term staurosporine treatment to induce apoptosis for the repeated CD40L blockage experiments (revised Figure 7).